# Effects of *Ficus carica* L. Water Extract on *Taxus cuspidata* Sieb. *et* Zucc. Growth

**Qianqian Li** [1,2,3,4,†], **Jin Huang** [1,2,3,4,†], **Xue Yang** [1,2,3,4], **Zarmina Gul** [1,2,3,4], **Wenxue Sun** [1,2,3,4], **Bin Qiao** [1,2,3,4],
**Jiabo Cheng** [1,2,3,4], **Chunying Li** [1,2,3,4,*] **and Chunjian Zhao** [1,2,3,4]

1   Key Laboratory of Forest Plant Ecology, Ministry of Education, Northeast Forestry University,
    Harbin 150040, China; lqq21@nefu.edu.cn (Q.L.); huangjin@nefu.edu.cn (J.H.); klp20yx@nefu.edu.cn (X.Y.);
    zarminagul@nefu.edu.cn (Z.G.); klp22swx@nefu.edu.cn (W.S.); klp20qb@nefu.edu.cn (B.Q.);
    ccwj11@nefu.edu.cn (J.C.); zcj@nefu.edu.cn (C.Z.)
2   College of Chemistry, Chemical Engineering and Resource Utilization, Northeast Forestry University,
    Harbin 150040, China
3   Engineering Research Center of Forest Bio-Preparation, Ministry of Education, Northeast Forestry University,
    Harbin 150040, China
4   Heilongjiang Provincial Key Laboratory of Ecological Utilization of Forestry-Based Active Substances,
    Harbin 150040, China
*   Correspondence: lcy@nefu.edu.cn; Tel./Fax: +86-451-8219-0848
†   These authors contributed equally to this work.

**Abstract:** Our research group successfully designed the *F. carica* and *Taxus cuspidata* Sieb. *et* Zucc mixed forest, and confirmed that their interspecific relationship was stable and *F. carica* has a promoting effect on neighboring *T. cuspidata* growth. However, the promoting mechanism has not been elucidated. In this study, *F. carica* was used as the donor plant and *T. cuspidata* was used as the recipient plant. *T. cuspidata* seedlings were irrigated with *F. carica* root extracts of different concentrations (10.0, 20.0, 40.0 g·L$^{-1}$), and the plant height, base diameter, photosynthetic parameters, photosynthetic pigments, MDA contents, and antioxidant enzyme activities were measured. Soil physical and chemical properties, enzyme activities, and microbial diversity were measured. The results showed that the abundance of growth-promoting bacteria increased and the number of pathogenic bacteria decreased in the rhizosphere of *T. cuspidata* soil. It was speculated that the chemicals secreted by *F. carica* roots interacted with soil microorganisms of *T. cuspidata* soil after enrichment, changed soil microbial diversity, and indirectly promoted the growth of *T. cuspidata*. UPLC-QTOF-MS/MS was used to analyze *F. carica* root water extract and *F. carica* root exudates, respectively, and it was found that the main components were similar. Therefore, the promoting effect of *F. carica* on *T. cuspidata* is mainly caused by the accumulation of potential chemicals in *F. carica* root exudates in the soil through interaction with soil microorganisms. Furthermore, the allelopathic-promoting mechanism of *F. carica* on *T. cuspidata* was discussed from various aspects, to provide a theoretical basis for the protection, breeding, and sustainable management of *T. cuspidata* resources.

**Keywords:** chemicals; irrigation; microbiology; promoting effect; soil

## 1. Introduction

*Taxus cuspidata* Sieb. *et* Zucc. is a small conifer or shrub belonging to the genus *Taxus*, family Taxaceae. In China, it is naturally distributed in the Northeast region and cultivated in Shandong, Jiangsu, and Xinjiang. Taxol, a new anticancer drug material that can effectively inhibit the growth of cancer cells, can be isolated from the roots, stems, and branches of *T. cuspidata* [1]. At present, most of the raw materials of taxol preparations are obtained from *T. cuspidata*, having a great demand in medicinal companies. In addition to its breeding characteristics, habitat conditions, climate change, and land use, it is now in the endangered state; it is listed as one of the plant species that is recognized as a national class I key protection of endangered plant species [2]. Mixed forest is a stable and efficient

artificial complex ecosystem that can promote plant development, and is thus a practical means of addressing the issue of *T. cuspidata*'s threatened status. Therefore, it is necessary to find a suitable tree species to mix with *T. cuspidata* to improve the survival rate and promote the growth of this plant.

*Ficus carica* Linn., belonging to the genus *Ficus*, family Moraceae, is one of the best medicinal and edible homologous tree species in the history of cultivation [3]. Primarily, these are small trees or deciduous shrubs; dioecious; with blooms enclosed in capsule-shaped receptacles; a fruit-like appearance; and no flowers. *F. carica* grows rapidly, having no strict requirements for its growth and soil conditions [4]. They are widely cultivated in China, and mainly distributed in Shandong, Xinjiang, Jiangsu, and Yunnan.

From April to September 2020, based on the characteristics of *T. cuspidata* with a non-fast-growing, sciophilous, medium–deep root system and *F. carica* with a fast-growing, heliophilous, shallow root system, our research team successfully designed a mixed forest of *F. carica* and *T. cuspidata* in Xiazhuang Experimental Field, Rongcheng City, Shandong Province, China (37°23′ N; 122°52′ E). The results showed that the interspecific relationship between *T. cuspidata* and *F. carica* species was stable [5]. *F. carica* can not only reduce the damage of strong light to *T. cuspidata* in its early growth period, saving the cost of planting *T. cuspidata* in the conventional shade shed, but also promote the growth of *T. cuspidata*. Compared with the pure forest of *T. cuspidata*, the plant height growth of *T. cuspidata* in the mixed forest increased by 22.33%, and the base diameter increased by 19.85%. However, the mechanism of *F. carica* promoting the growth of *T. cuspidata* has not been clarified.

The concept of allelopathy (chemical interaction between species) was first proposed by Hans Molisch, and its connotation has been continuously enriched by subsequent scholars [6]. The broad sense of allelopathy includes one plant causing both the promotion and inhibition of the growth of other neighboring plants [7,8]. In the process of growth and development, plants release specific metabolites to the environment, change the surrounding microecological environment, and affect the growth and development of other plants around, and this leads to mutual exclusion or promotion among plants [9,10]. Rational use of positive allelopathy can achieve the dual purpose of improving forest productivity and maintaining ecosystem stability [11]. Therefore, it is of great practical significance to reveal the interspecific allelopathy mechanism of mixed forest ecosystems to fully understand the interspecific relationships of specific combination trees and then to build mixed forests with coordinated interspecific relationships.

The chemicals released into the environment by plants can affect the chlorophyll contents of receptor plant cells by changing the formation of photosynthetic pigments, the light reaction and dark reaction in plants, the process of chlorophyll synthesis, and the photosynthetic rate of plants, thus affecting the photosynthesis of plants [12]. The activity levels of superoxide dismutase (SOD), peroxidase (POD), and catalase (CAT) can reflect the antioxidant capacity of the body and are the most important factors of the active oxygen scavenging system in plants. They interact and work together to effectively prevent the harm that reactive-oxygen-species-induced membrane lipid peroxidation causes to cells [13]. Malondialdehyde (MDA) content indirectly reflects the degree of tissue peroxidation damage and is an important parameter to indirectly judge the stress resistance of plants and the degree of cell membrane damage [14]. Soil physical and chemical properties, soil enzyme activities, and soil microorganisms are all stable indicators to evaluate soil quality. They are not only important indicators of soil fertility but can also reflect the maintenance of plant growth conditions, plant stress resistance, and the intensity of soil metabolism to a certain extent, and play an important role in the energy flow and material cycle of terrestrial ecosystems [15–17].

In this study, we used water extract of *F. carica* to irrigate seedlings of *T. cuspidata* in a pot experiment and measured the growth and physiological indicators (plant height, basal diameter, gas exchange parameters, chlorophyll contents, antioxidant enzymes, and MDA) and soil properties (soil physical and chemical property, soil enzyme activity, and soil microbial diversity) of *T. cuspidata*. Furthermore, the potential functional chemicals

were screened out. The aim is to explore the specific mechanism of *F. carica* promoting the growth of *T. cuspidata* and the potential growth-promoting chemicals through allelopathy between plants, which is helpful to solve the problem of the slow growth of *T. cuspidata* seedlings. This study provides new insights into the effective and sustainable management of an artificial *F. carica* and *T. cuspidata* mixed forest in an agroforestry ecological system.

## 2. Materials and Methods

### 2.1. Plant Materials and Reagents

The donor plant *F. carica* and the recipient plant *T. cuspidata* were collected in Chengshan Town, Rongcheng City, Shandong Province.

The ultrapure water used in the experiment was prepared by a Standard Unique-LC R20 ultrapure water machine (Xiamen Research Water Purification Technology Co., Ltd., Fujian, China). Mass spectrum grade methanol, acetonitrile and formic acid were purchased from Thermo Fisher Scientific (Waltham, MA, USA). Pure aquatic products for UPLC-MS/MS analysis from Wahaha Group Co., LTD. (Hangzhou, Zhejiang, China). Other chemicals and solvents were analytical grade and purchased from Tianjin Comer Chemical Reagent Company (Tianjin, China).

### 2.2. Effects of Extracts from Different Parts of F. carica on the Growth of T. cuspidata

2.2.1. Sample Collection and Processing

The pot experiment was conducted in the Key Laboratory of Forest Plant Ecology of the Ministry of Education (45°43′8″ N, 126°38′3″ E) at Northeast Forestry University for 180 days from 15 March to 15 September 2022. The temperature of the greenhouse was kept at 15~30 °C, the transmittance was about 75% of the natural illumination, the air humidity was controlled at 50%–70%, the daily light time was 12~14 h, and the appropriate weather was regularly selected for ventilation. The soil used in the experiment was taken from the topsoil (0–20 cm) of the Xiazhuang experimental field in Chengshan Town, Rongcheng City, Shandong Province. The upper mouth diameter of the cultivation pots was 20 cm, the lower mouth diameter was 18 cm, the depth of the pot was 25 cm, and the soil in each pot was 6 kg. The healthy fresh roots, stems, and leaves of *F. carica* without infection of diseases and pests were collected. After cleaning the surrounding soil, it was rinsed with water and placed on dry, clean filter paper to drain. After the moisture on the surface was completely dry, it was placed in the oven with the temperature set at 60 °C. After drying to a constant weight, it was smashed with a shredder and passed through a 2 mm sieve.

The root exudate solution was obtained by hydroponics [18]. A potted *F. carica* seedling with good growth was selected from the greenhouse of Northeast Forestry University. The roots were soaked in deionized water for 5 min, during which time the roots were gently shaken every 1 min to ensure that the soil and impurities were washed. The *F. carica* plant was then cultured at 25 ± 1 °C for 7 days in a cylindrical glass bottle (14 mm diameter and 86 mm height) containing 100 mL of highly purified water and the photocycle was 12 h. Deionized water was added every 24 h to keep the solution in the container at the same level. Only seedling roots were immersed in the solution during incubation. The high pure water culture medium in a glass bottle was the solution containing root exudates. After vacuum freeze-drying, the sample was ground with a grinder (30 Hz, 1.5 min) to powder form. An amount of 100 mg powder was weighed and dissolved in 1.2 mL 70% methanol solution. To improve the extraction rate, the samples were vortexed every 30 min for 30 s and 6 times in total. The samples were placed in a 4 °C refrigerator overnight. Finally, the samples were centrifuged (12,000 r·min$^{-1}$, 10 min), the supernatant was absorbed, and the samples were filtered by a microporous filter membrane (pore size 0.22 μm), and stored in the sample bottle for LC-MS/MS analysis.

2.2.2. Preparation of Water Extracts from Different Parts of *F. carica*

After accurately weighing 20 g of *F. carica* roots, stems, and leaves powder, ultrasonic extraction for 30 min, and centrifugation for 10 min at 12,000 r·min$^{-1}$, the supernatant

was taken to obtain the extract solution with a concentration of 40.0 g·L$^{-1}$, which was diluted into 20.0 g·L$^{-1}$ and 10.0 g·L$^{-1}$ solutions, respectively. The above three concentrations of extracting solution were stored in the refrigerator at 4 °C for subsequent pot watering experiments.

### 2.2.3. Pot Experiment of *T. cuspidata* Irrigated with Water Extracts from Different Parts of *F. carica*

There were three treatments: *F. carica* root extract, *F. carica* stem extract, and *F. carica* leaf extract. Distilled water was used as the control, and each treatment had 3 replicates. *T. cuspidata* was irrigated thoroughly with *F. carica* extract from different parts for the first time, and then irrigated once every 7 days with 100 mL of the extract in each pot, and 100 mL of distilled water was added to the control. After 30 days of irrigation, the plant height and base diameter of *T. cuspidata* were measured using a tape measure and vernier caliper, respectively, and the average value of 3 measurements was taken as the growth index.

### *2.3. Effects of F. carica Root Water Extract on the Growth of T. cuspidata*
### 2.3.1. Preparation of *F. carica* Root Water Extract

The *F. carica* root powder 5.0 g was accurately weighed and placed in a triangle bottle with 100 mL distilled water and extracted at room temperature for 24 h, during which it was shaken every 15 min, and then ultrasonic for 30 min. The residue was filtered with four layers of gauze and collected into a centrifuge tube. The supernatant was centrifuged for 10 min at 12,000 r·min$^{-1}$. A total of 40.0 g·L$^{-1}$ fresh water extract was prepared and diluted to a concentration of 20.0 g·L$^{-1}$ and 10.0 g·L$^{-1}$, and then used in a refrigerator at 4 °C.

### 2.3.2. Irrigation of *T. cuspidata* with Water Extract of *F. carica* Root

There were three treatments: *F. carica* root extract of 10.0 g·L$^{-1}$, 20.0 g·L$^{-1}$, 40.0 g·L$^{-1}$, and distilled water as control. Three replicates were set for each treatment concentration, and the average values of 3 measurements were taken as the results of each growth index. *T. cuspidata* was irrigated with different concentrations of *F. carica* root extract thoroughly for the first time, and then irrigated once every 7 days for each pot, and the control was irrigated with distilled water. Each treatment was irrigated for 180 days. Growth, physiological indices, and soil property were measured on 15 April (30 days' irrigation), 1 July (105 days' irrigation), and 15 September (180 days' irrigation), 2022.

### 2.3.3. Determination of Plant Height and Base Diameter of *T. cuspidata*

The plant height was measured with a tape measure (units: cm; precision: 0.1 cm), and the basal diameter was measured with a vernier caliper (units: mm; precision: 0.02 mm) (Pulisen Measuring Tools Co., Ltd., Harbin, China).

### 2.3.4. Determination of Gas Exchange Parameters

The LI-6400 portable photosynthesis system (LiCor, Lincoln, NE, USA) was used to measure the gas exchange parameters of leaves at 9:00 a.m. on a sunny day [19]. The mature leaves of the apex with good growth and brightness were selected, with 3 replicates for each treatment. The concentration of $CO_2$ was 452 μmol·mol$^{-1}$, and the saturation light intensity was 1000 μmol·m$^{-2}$·s$^{-1}$ in the greenhouse. Net photosynthetic rate (Pn, μmol·m$^{-2}$·s$^{-1}$), stomatal conductance (Gs, mmol·m$^{-2}$·s$^{-1}$), transpiration rate (Tr, mmol·m$^{-2}$·s$^{-1}$), and intercellular $CO_2$ concentration (Ci, μmol·mol$^{-1}$) were measured.

### 2.3.5. Determination of Chlorophyll Contents

The chlorophyll contents of plants were determined by UV spectrophotometry (UV-2600, Shimadzu Instruments Co., Ltd., Suzhou, China). The mixed solution of acetone–absolute ethanol =1:1 was taken as the blank control, and the absorbance measured at

the wavelengths of 645 nm and 663 nm, respectively [20,21]. The concentration of $C_a$, $C_b$, and $C_T$ (i.e., chlorophyll a, b, and total chlorophyll, mg·L$^{-1}$) was calculated by the following formula.

C: Chlorophyll concentration (mg·L$^{-1}$);
V: Total extract volume (mL);
M: Fresh leaf weight (g).

### 2.3.6. Determination of Protective Enzyme Activity and MDA Contents

Antioxidant enzyme activity and MDA content were determined according to Abdelkader et al. [22].

### 2.4. Effects of F. carica Root Extract on Soil Properties of T. cuspidata

#### 2.4.1. Collection of Soil Samples

After 30 days (15 April), 105 days (1 July), and 180 days (15 September) of pot experiment irrigation, S-type sampling was carried out for each treatment group, and each treatment was repeated 3 times. The collected soil sample was placed into the numbered sterile sealed bag and the sample brought back to the laboratory as soon as possible. The soil samples obtained were uniformly mixed and then passed through a 2 mm soil sieve, which was equally divided into 3 parts. The first part was dried naturally and used for soil nutrient contents determination, the second part was stored at 4 °C for soil enzymatic activity determination, and the third part was stored at −80 °C for soil microbial diversity determination.

#### 2.4.2. Measurement of Soil Physicochemical Property

The physical and chemical properties of soil were determined by Huang et al. [23] and Raya-Moreno et al. [24]. Soil pH was measured with a pH meter (Sartorius PT-21, Beijing Ouxinsheng Technology Co., Ltd., Beijing, China). Soil electrical conductivity (EC) was measured with an electrical conductivity meter (DDS-11A, Shanghai INESA Scientific Instrument Co., Ltd., Shanghai, China). Soil organic carbon (SOC) was determined by the external heating method of potassium dichromate oxidation. Soil available nitrogen (AN) was determined by alkaline hydrolysis diffusion method. Available phosphorus (AP) in soil was determined by the molybdenum antimony colorimetric method. Available potassium (AK) in soil was determined by the flame photometric method.

#### 2.4.3. Determination of Soil Enzyme Activity

The activities of urease, catalase, and sucrase in soil were determined according to Zhang et al. [25]. Cellulase activity was determined according to the method of Shen et al. [26].

#### 2.4.4. Determination of Soil Microbial Diversity

Illumina Hiseq sequencing method was used to analyze the diversity of soil bacteria and fungi by DNA extraction, PCR amplification, fluorescence quantification, Miseq library construction, and Miseq sequencing.

### 2.5. Identification of Potential Chemicals in Water Extract of T. cuspidata

The potential chemicals in the water extract of *T. cuspidata* were identified by UPLC-MS/MS (Dionex UIiMate 300, Themo Scientific, Waltham, MA, USA). The chromatographic column used was the Hyperil Gold column (100 × 2.1 mm, 1.9 μm). In the positive ion mode, gradient elution was used to achieve the separation of compounds; the mobile phase under this elution was 0.1% formic acid-$H_2O$ (solvent A) and methanol (solvent B). The mobile phase under the negative ion mode was 5 mM ammonium acetate-$H_2O$ (pH = 9, ammonia water conditioning, solvent A) and methanol (solvent B). The elution conditions under positive and negative ions were the same: 5% B (0–0.45 min), 5–100% B

(0.45–30.05 min), 100% B (30.05–34.27 min), 100–5% B (34.27–34.70 min), and 3.8 min were used to balance the initial conditions, with an analysis time of 38.5 min.

### 2.6. Statistical Analysis

Xcalibur and Compound Discoverer 3.1 software were used to analyze the MS data. All data were analyzed by SPSS22.0 statistical software, and one-way ANOVA and Duncan's Multiple-range Tests ($p < 0.05$) were used to further deal with data differences. Origin2021 was used for mapping. The microbial species composition, diversity, and richness were analyzed using the Majorbio cloud platform online platform at www.majorbio.com (accessed on 29 November 2022).

## 3. Results

### 3.1. Effects of Extracts from Different Parts of F. carica on the Growth of T. cuspidata

The changes in plant height and base diameter of *T. cuspidata* irrigated by different parts (root, stem, and leaf) of *F. carica* were measured. As shown in Figure 1a, *T. cuspidata's* plant height increased slowly after being irrigated with different concentrations of *F. carica* stem and leaf extract, and there was no significant difference between them and the control ($p > 0.05$). Only 10.0 g·L$^{-1}$ and 20.0 g·L$^{-1}$ concentrations of extracts showed a promoting effect on the plant height of *T. cuspidata*. The plant height of *T. cuspidata* irrigated with different concentrations of *F. carica* root extract was higher than that of the control, and there were significant differences between the two groups ($p < 0.05$). The plant height of *T. cuspidata* reached the highest value 24.40 cm by irrigating with the concentration of 10.0 g·L$^{-1}$ *F. carica* root extract, and significantly increased by 15.80% compared with the control ($p < 0.05$).

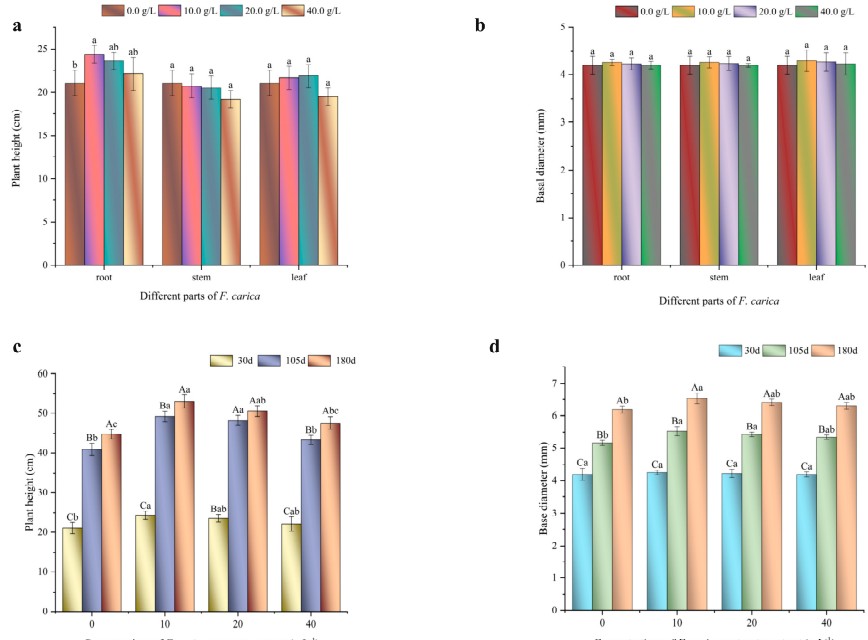

**Figure 1.** Plant height (**a**) and base diameter (**b**) of *T. cuspidata* irrigated with water extract from different parts (root, stem, and leaf) of *F. carica*. Plant height (**c**) and base diameter (**d**) of *T. cuspidata* irrigated with water extract of *F. carica* root. Values are reported as mean ± SD, n = 3. Different uppercase letters in the different concentrations indicate significant differences at $p < 0.05$ among different irrigated times; different lowercase letters in the different irrigation times indicate significant differences at $p < 0.05$ among different concentrations irrigated.

According to Figure 1b, the basal diameter of *T. cuspidata* irrigated with different concentrations of *F. carica* root extract was higher than that of the control, but there was no

significant difference ($p > 0.05$). The *F. carica* stem extract with a concentration of 40.0 g·L$^{-1}$ showed an inhibitory effect on the basal diameter of *T. cuspidata*. The basal diameter of *T. cuspidata* irrigated with *F. carica* leaf extract was higher than that of the control only at the concentration of 10.0 g·L$^{-1}$ but was lower than that of the control under other concentrations irrigated.

It can be concluded that the extracts from different parts of a plant have different effects on the same receptor plants. In this test, *F. carica* root extract had a better-promoting effect on the growth of *T. cuspidata* than other parts (stem and leaf), so the *F. carica* root extract with a stronger promoting effect was selected for subsequent irrigation in the experiment.

### 3.2. Effect of F. carica Water Extract on the Growth of T. cuspidata

3.2.1. Effect of *F. carica* Water Extract on the Plant Height and Base Diameter of *T. cuspidata*

The plant morphology index can directly reflect the influence of water extract of some donor plants on the recipient plants, and plant height and base diameter are the most intuitive expression of its growth [27]. The effects of different concentrations of *F. carica* root water extract on plant height and base diameter of *T. cuspidata* were measured. The plant height and base diameter of *T. cuspidata* irrigated with the same concentration of *F. carica* root extract increased gradually with the increase in irrigating time. As shown in Figure 1c, the plant height of *T. cuspidata* irrigated with 10.0 g·L$^{-1}$ *F. carica* root extract increased rapidly. After 30 and 105 days of irrigation, the plant height of *T. cuspidata* was increased by 15.82% and 20.46% compared with the control. At 180 days of irrigating, the maximum value of 53.03 cm was achieved, which increased by 18.29% compared with the control. When irrigated with 40.0 g·L$^{-1}$ *F. carica* root extract for 30 days, the plant height of *T. cuspidata* irrigated was 1.05 times that of the control, and was not significantly increased at 105 days of irrigation compared with the control ($p > 0.05$); it only increased by 5.94%. After 180 days of irrigation, it increased by 6.34% compared with the control and the difference was significant ($p < 0.05$).

According to Figure 1d, the basal diameter of *T. cuspidata* irrigated with 10.0 g·L$^{-1}$ *F. carica* root extract increased rapidly by 1.43% compared with the control at 30 days of irrigating. After 105 days of irrigation, it increased by 7.17% compared with the control and the difference was significant ($p < 0.05$). At 180 days, the maximum value of 6.53 mm was reached, which was significantly increased by 5.66% compared with the control. On the other hand, the basal diameter of *T. cuspidata* irrigated with 40.0 g·L$^{-1}$ *F. carica* root extract increased slowly, and there was almost no change in basal diameter after 30 days of irrigating, and the promoting effect was not significant ($p > 0.05$). The basal diameter of *T. cuspidata* irrigation with 105 days increased by 3.49% compared with the control and the difference was significant ($p < 0.05$). After 180 days of irrigating, the basal diameter was 1.06 times that of the control, significantly increased by 1.94% ($p < 0.05$).

3.2.2. Effects of *F. carica* Water Extract on Gas Exchange Parameters of *T. cuspidata*

Photosynthesis is an important biological process of plant life, and plays an important role in the improvement of the earth's ecosystem. The metabolic energy of higher plants comes from photosynthesis, through which the energy of sunlight is captured and converted into biological energy [28,29]. Figure 2 shows the results of Pn, Tr, Gs, and Ci of *T. cuspidata* irrigated by *F. carica* root extract with different concentrations. Under the irrigation of *F. carica* root extract, the contents of Pn, Tr, Gs, and Ci in *T. cuspidata* continued to increase with the increase in irrigating time, and there were significant differences compared to the control ($p < 0.05$). Under lower concentrations (10.0 g·L$^{-1}$ and 20.0 g·L$^{-1}$) of *F. carica* root extract, the contents of Pn, Tr, Gs, and Ci were significantly higher than those in the control ($p < 0.05$). However, under high concentrations (40.0 g·L$^{-1}$) of *F. carica* root extract, the contents of Pn, Tr, Gs, and Ci were lower than those in the control.

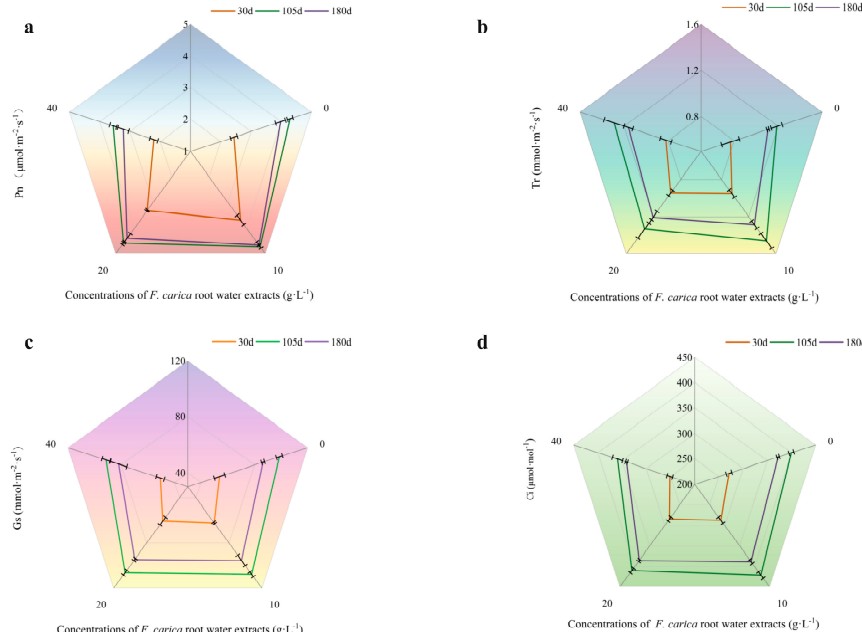

**Figure 2.** The contents of Pn (**a**), Tr (**b**), Gs (**c**), and Ci (**d**) of *T. cuspidata* irrigated with water extract of *F. carica* root. Values are reported as mean ± SD, n = 3.

The highest Pn value was 4.72 μmol·m$^2$·s$^{-1}$ after 105 days of irrigation with 10.0 g·L$^{-1}$ *F. carica* root extract, which was an increase of 9.94% compared with the control. The content of Pn of *F. carica* root extract irrigated with a high concentration of 40.0 g·L$^{-1}$ was lower than that of the control group, showing an inhibitory effect. At 30 days of irrigation, the difference was not significant ($p > 0.05$), and at 105 and 180 days of irrigation, the difference was significantly decreased by 17.24% and 19.03% compared with the control (Figure 2a). The highest Tr value was 1.46 μmol·m$^2$·s$^{-1}$ after 105 days of 10.0 g·L$^{-1}$ *F. carica* root extract irrigation, which was an increase of 23.03% compared with the control. Although the Tr in the *F. carica* root extract irrigation with a high concentration of 40.0 g·L$^{-1}$ was still higher than that in the control, the promotion effect was weak, and it only increased by 6.96%, 8.71%, and 4.83% compared with the control at 30 days, 105 days, and 180 days, respectively (Figure 2b). The lowest content of Gs was 50.63 μmol·m$^2$·s$^{-1}$ when the concentration of *F. carica* root extract was 40.0 g·L$^{-1}$ for 30 days, which was 6.00% lower than that of the control ($p > 0.05$). The highest value was 107.57 μmol·m$^2$·s$^{-1}$, at the concentration of 10 g·L$^{-1}$ *F. carica* root extract for 105 days, which was a significant increase of 8.87% compared with the control ($p < 0.05$) (Figure 2c). The highest Ci content was 421.00 μmol·mol$^{-1}$ when the concentration of 10.0 g·L$^{-1}$ *F. carica* root extract was irrigated for 180 days, which was an increase of 5.34% compared with the control. The content of Pn in the *F. carica* root extract irrigation group with a high concentration of 40.0 g·L$^{-1}$ was lower than that in the control, and showed a significant difference ($p < 0.05$). The contents of Ci in the *F. carica* root extract irrigation were reduced by 7.26%, 10.10%, and 8.51% compared with the control at 30 days, 105 days, and 180 days, respectively (Figure 2d).

### 3.2.3. Effect of *F. carica* Water Extract on Chlorophyll Contents of *T. cuspidata*

Chlorophyll is the basis of plant photosynthesis, which can directly affect the intensity of photosynthesis, photosynthetic efficiency, and the accumulation of photosynthetic products in plants, and play a key role in plant day lighting and light protection [30]. The changes in chlorophyll contents (chlorophyll a, chlorophyll b, and total chlorophyll) of *T. cuspidata* were measured by irrigating different concentrations of *F. carica* root extracts. Under the same irrigation concentration, chlorophyll a contents in *T. cuspidata* leaves showed a trend of first increasing and then decreasing with the increase in irrigation time, and all groups reached the maximum value at 105 days of irrigation. *F. carica* root extracts

with 10.0 g·L$^{-1}$ and 20.0 g·L$^{-1}$ irrigation were 1.28 and 1.26 times that of the control, respectively. Compared with the control, *F. carica* root extract at the concentration of 40.0 g·L$^{-1}$ was reduced by 16.07% (Figure 3a). The chlorophyll b content in the leaves of *T. cuspidata* irrigated with 10.0 g·L$^{-1}$ *F. carica* root extract was significantly different from that of the control ($p < 0.05$), and increased by 19.87%, 32.62%, and 25.02% at 30, 105, and 180 days, respectively. At 105 days of irrigation, the maximum value was 6.36, 1.33 times that of the control (Figure 3b). Total chlorophyll contents also showed a trend of first increase and then decrease with the increase in irrigation time, and all groups reached their maximum value at 105 days of irrigation. When 10.0 g·L$^{-1}$ *F. carica* root extract was irrigated for 105 days, the maximum value was 20.48 mg·g$^{-1}$, which was 1.03 times that of the control and the difference was significant ($p < 0.05$). The *F. carica* root water extract at 40.0 g·L$^{-1}$ was reduced by 16.03% compared with the control (Figure 3c).

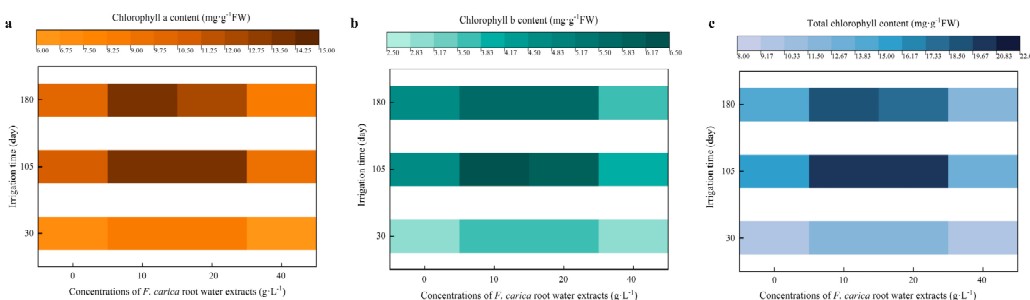

**Figure 3.** The contents of chlorophyll a (**a**), chlorophyll b (**b**), and total chlorophyll (**c**) of *T. cuspidata* irrigated with water extract of *F. carica* root. Values are reported as mean ± SD, n = 3.

### 3.2.4. Effects of *F. carica* Water Extract on Enzymatic Activity and Malondialdehyde Contents of *T. cuspidata*

When plants are in a stressed environment, SOD, POD, and CAT in the antioxidant protection enzyme system of plants can work together to eliminate $O_2^-$, $H_2O_2$, and other oxygen free radicals produced during the growth of plants [31]. The enhancement of their activity can effectively prevent the accumulation of reactive oxygen species and reduce the damage to plant cell membranes, to ensure that MDA is maintained at a low level [32]. It plays an important role in maintaining the normal growth of plants. The activity of three main protective enzymes (SOD, POD, CAT) and the contents of MDA in *T. cuspidata* irrigated with *F. carica* root water extract at different concentrations was measured. The activity of SOD, POD, and CAT in *T. cuspidata* increased when the concentration of *F. carica* root extract was 10.0–20.0 g·L$^{-1}$, and the increase in activity was most obvious when the concentration was 10.0 g·L$^{-1}$. The variation range of SOD was 307.64–652.38 U·g$^{-1}$·min$^{-1}$ FW. After 105 days of irrigation, SOD activity was significantly increased by 27.24% ($p < 0.05$) to the maximum. When the concentration was 40.0 g·L$^{-1}$, the activity of SOD decreased by 14.80%, 15.65%, and 17.00% at 30, 105, and 180 days, respectively, compared with the control ($p > 0.05$) (Figure 4a). POD ranged from 396.45–740.35 U·g$^{-1}$·min$^{-1}$ FW, and reached the maximum value at 10.0 g·L$^{-1}$ *F. carica* root extract for 105 days, which was a significant increase of 38.89% compared with the control ($p < 0.05$). However, the promoting effect started to decrease when the concentration was 40.0 g·L$^{-1}$, and the POD activity was only 1.03 times that of the control at 180 days of irrigation. There was no significant difference ($p > 0.05$) (Figure 4b). CAT ranged from 447.13 to 707.18 U·g$^{-1}$·min$^{-1}$ FW. When the concentration was 10.0 g·L$^{-1}$, the promoting effect was most obvious. Compared with the control, CAT activity was increased by 29.05% and the difference was significant ($p < 0.05$) at 105 days, reaching the maximum. However, when the concentration was 40.0 g·L$^{-1}$, the promoting effect began to decrease, although it was still higher than the control, and the CAT activity was only 1.05 times that of the control after 180 days of irrigation (Figure 4c).

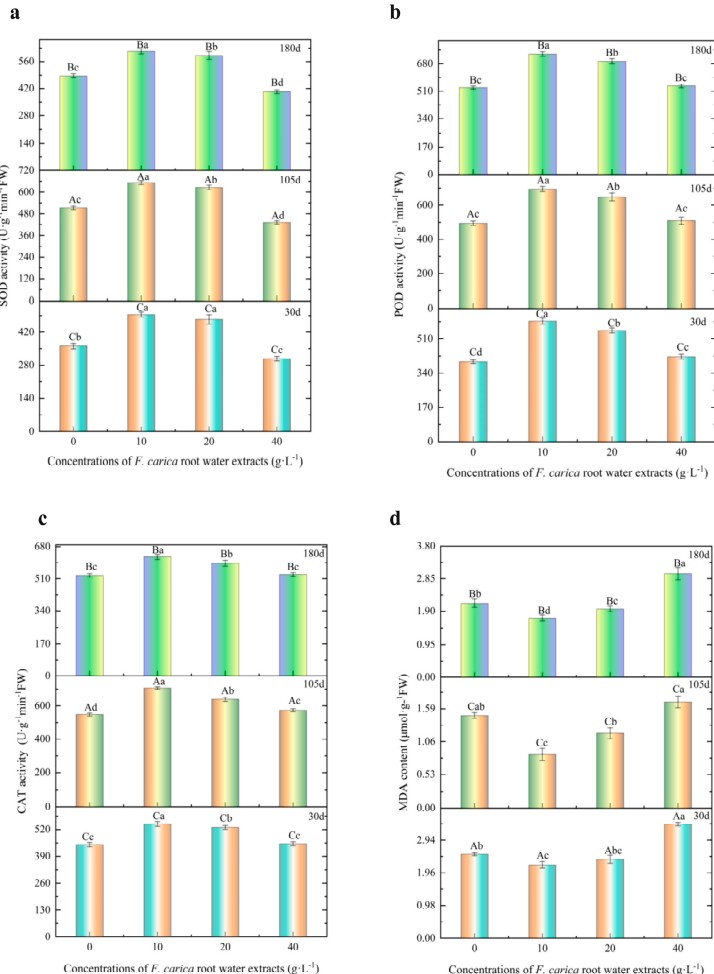

**Figure 4.** SOD (**a**), POD (**b**), CAT (**c**) activity and MDA (**d**) contents of *T. cuspidata* irrigated with water extract of *F. carica* root. Values are reported as mean ± SD, n = 3. Different uppercase letters in the same row indicated significant differences at *p* < 0.05 among different treatments time; Different lowercase letters in the same column indi-cated significant differences at *p* < 0.05 among different concentration treatments.

MDA contents ranged from 0.85 to 3.42 μmol·g$^{-1}$ FW. The influence of different concentrations of *F. carica* root extract on the MDA content of *T. cuspidata* was inhibited. The MDA contents in the low concentration (10.0 g·L$^{-1}$) and medium concentration (20.0 g·L$^{-1}$) *F. carica* root extract irrigation were always significantly lower than that in the control (*p* > 0.05), and reached the lowest value at 105 days of irrigation. They were 42.49% and 19.31% lower than the control, respectively. The content of MDA in 40.0 g·L$^{-1}$ *F. carica* root extract irrigation was significantly higher than that in the control (*p* < 0.05), and reached the maximum value at 30 days of irrigation, which was 38.14% higher than that in the control. The inhibitory effect was strongest at 10.0 g/L and reached 0.85 μmol·g-1FW at 105 days, while the MDA content in the control group was 1.74 times that of this group. In addition, the concentration of *F. carica* root extract was lower than that of other groups at different irrigation periods, and the inhibition effect of *T. cuspidata* roots was the least, and the damage to the antioxidant system of the roots of *T. cuspidata* was also the least (Figure 4d).

### 3.3. Effects of F. carica Water Extract on Soil Properties of T. cuspidata

3.3.1. Changes in Soil Physicochemical Properties

Many studies have shown that the contents of soil pH, soil organic matter, total nitrogen, total phosphorus, ammonium nitrogen, nitrate nitrogen, available phosphorus,

total potassium, and available potassium vary with different concentrations of donor plant water extracts, which changes soil physical and chemical properties and soil quality, thus indirectly affecting the growth and development of recipient plants [33]. We tested the effects of different concentrations of *F. carica* root extracts on soil physicochemical properties (pH, EC, SOC, AN, AP, AK) of *T. cuspidata*.

With the increase in the concentration of *F. carica* root extract, the soil pH of *T. cuspidata* showed a trend of first rising to a certain value and then decreasing, but the overall value remained above the control group. After 180 days of irrigating, the soil pH values of *T. cuspidata* with 10.0 $g·L^{-1}$, 20.0 $g·L^{-1}$, and 40.0 $g·L^{-1}$ *F. carica* root extract were significantly different from that of the control ($p < 0.05$), and were 1.92%, 1.01%, and 0.69% higher than that of the control, respectively (Figure 5a). On the whole, different concentrations of *F. carica* root extracts could improve the pH of *T. cuspidata* soil, which was in the range of 6.07–6.41, belonging to acidic soil (pH 5.0–6.5). The results indicated that the *F. carica* root extract might contain antibacterial substances, which could improve the microecological environment and physical and chemical properties of *T. cuspidata* soil and promote the growth and development of plants. It may also be that *F. carica* root extract has a strong ability to absorb inorganic nitrogen, which increases the soil pH of *T. cuspidata*, changes soil physical and chemical properties, and maintains soil fertility [34]. The lowest EC was found in *F. carica* root extract with a concentration of 20.0 $g·L^{-1}$ for 30 days, which could significantly reduce the soil EC of *T. cuspidata* ($p < 0.05$) and was 5.80% lower than that of the control. After 180 days of irrigation, all three irrigation concentrations (10.0 $g·L^{-1}$, 20.0 $g·L^{-1}$, and 40.0 $g·L^{-1}$) could significantly reduce the soil EC of *T. cuspidata* ($p < 0.05$), and the lower the concentration was, the lower the soil EC was. They were 4.04%, 5.60%, and 3.84% lower than the control, respectively. On the whole, different concentrations of *F. carica* root extracts could reduce the soil EC of *T. cuspidata*, and the EC ranged from 13.41 to 22.73 $ms·m^{-1}$. The soil EC of *T. cuspidata* was significantly decreased after different concentrations of *F. carica* root extract were irrigated, which showed a law of first increasing and then decreasing with the change in irrigation time, and reached the peak value at 105 days, which may be because the high temperature in July resulted in increased water evaporation, increased salt contents, and increased soil EC. Cold autumn weather in September resulted in less water evaporation, less salt, and lower soil EC [35]. With the increase in treatment time, the SOC contents showed a trend of first increasing and then decreasing, and reached the highest value at 105 days. However, the SOC content of *T. cuspidata* irrigated with high concentration (40.0 $g·L^{-1}$) *F. carica* root extract had no significant difference compared with that of the control, and only increased by 1.16%. The SOC contents of *T. cuspidata* irrigated with low concentration (10.0 $g·L^{-1}$) and medium concentration (20.0 $g·L^{-1}$) *F. carica* root extract were significantly higher than that of the control by 29.49% and 10.08% ($p < 0.05$), respectively. The *F. carica* root extract may accelerate the growth of *T. cuspidata* underground parts, increasing the number of root secretions, increasing the soil's carbon source, stimulating the soil microbial activity of *T. cuspidata*, accelerating the decomposition of soil organic matter, and producing more SOC [36].

According to Figure 5b, under the same irrigation time, with the increase in *F. carica* root water extract concentration, soil AN, AP, and AK contents all increased to a certain value and then decreased. A low concentration (10.0 $g·L^{-1}$) and medium concentration (20.0 $g·L^{-1}$) of *F. carica* root extract increased soil AN contents of *T. cuspidata*, and there were significant differences ($p < 0.05$). The soil AN content of *T. cuspidata* irrigated with 10.0 $g·L^{-1}$ *F. carica* root extract had the most significant difference compared with the control ($p < 0.05$), and increased by 16.30%, 17.87%, and 13.59% at 30 days, 105 days, and 180 days, respectively. Different concentrations of *F. carica* root extract increased the soil AP contents of *T. cuspidata*. There were significant differences in soil AP contents ($p < 0.05$), and the range of soil AP contents was 17.31–40.59 $mg·kg^{-1}$. The elevation effect was most obvious when the concentration was 10.0 $g·L^{-1}$, and the soil AP content of *T. cuspidata* was significantly increased by 69.13% ($p < 0.05$) at 105 days of irrigation, reaching the maximum 40.59 $mg·kg^{-1}$ compared with the control. A low concentration (10.0 $g·L^{-1}$)

and medium concentration (20.0 g·L$^{-1}$) of *F. carica* root extract increased soil AK contents of *T. cuspidata*, and there were significant differences in soil AK contents ($p < 0.05$); the range of soil AK contents was 76.24–153.13 mg·kg$^{-1}$. After 180 days of irrigation, the soil AK contents of *T. cuspidata* irrigated with the three concentrations of *F. carica* root extract reached their maximum values, which were 39.18%, 29.42%, and 2.15% higher than that of the control, respectively.

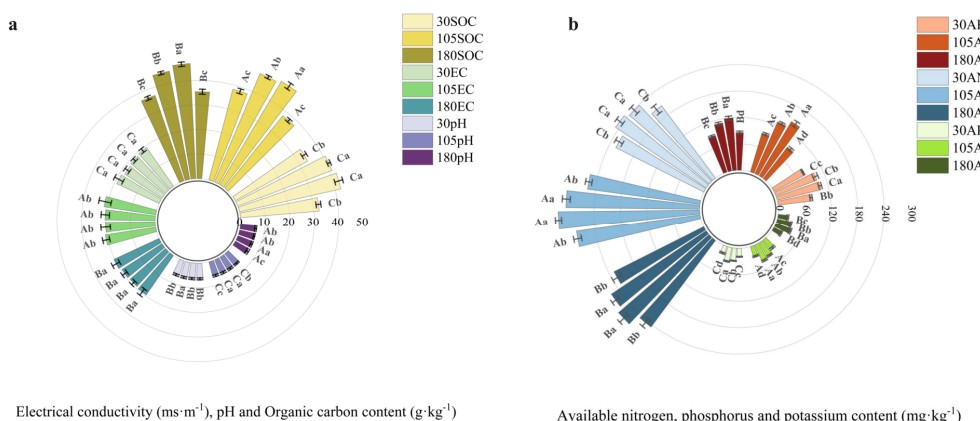

**Figure 5.** The contents of EC, SOC, and the pH (**a**), AN, AP, and AK contents (**b**) in *T. cuspidata* irrigated with water extract of *F. carica* root. Values are reported as mean ± SD, n = 3. Different uppercase letters in the same row indicated significant differences at $p < 0.05$ among different treatments time; Different lowercase letters in the same column indi-cated significant differences at $p < 0.05$ among different concentration treatments.

3.3.2. Changes in Soil Enzymatic Activities

Plant development and soil enzyme activity are closely related, and soil enzyme activity is a key measure of soil biological efficiency. It is often used to evaluate soil fertility, ripening degree, and fertility level, and is an indicator of biological activity indirectly reflecting soil quality [37]. We tested the activities of urease, catalase, cellulase, and sucrase in the soil of *T. cuspidata* irrigated with different concentrations of *F. carica* root extract. Under the same concentration of *F. carica* root water extract irrigation, the four soil enzyme activities showed an increase first and then a decrease with the change in irrigation time. The activities of urease, catalase, cellulase, and sucrase in the soil of *T. cuspidata* were 0.33–0.86 mg·g$^{-1}$, 0.23–0.84 mg·g$^{-1}$, 0.12–0.54 mg·g$^{-1}$, and 22.79–68.63 mg·g$^{-1}$, respectively.

A low concentration (10.0 g·L$^{-1}$) and medium concentration (20.0 g·L$^{-1}$) of *F. carica* root extract increased the activity of soil urease, and there were significant differences ($p < 0.05$). After 30 days of irrigation, it increased by 31.96% and 41.39%, respectively, compared with the control. The maximum value was reached at 105 days of irrigation, with increases of 36.95% and 42.99% compared with the control. After 180 days of irrigation, the difference was significant compared with the control ($p < 0.05$), and was 1.42 times and 1.48 times, respectively (Figure 6a). The maximum soil catalase value of 0.84 mg·g$^{-1}$ appeared at 10.0 g·L$^{-1}$ *F. carica* root water extract for 105 days, which was a significant increase of 70.59% compared with the control. The maximum values of 20.0 g·L$^{-1}$ and 40.0 g·L$^{-1}$ also appeared at 105 days of irrigation, with increases of 67.65% and 29.41% compared with the control, respectively (Figure 6b). At a low concentration (10.0 g·L$^{-1}$) and medium concentration (20.0 g·L$^{-1}$) of *F. carica* root extract, the soil cellulase activity of *T. cuspidata* was significantly increased ($p < 0.05$), and was 44.10% and 27.25% higher than that of the control at 30 days of irrigation, respectively. When 10.0 g·L$^{-1}$ was irrigated for 105 days, the maximum value was 57.13% higher than that of the control (Figure 6c). After 105 days of irrigating with 20.0 g·L$^{-1}$ *F. carica* root water extract, soil sucrase activity reached the maximum of 68.63 mg·g$^{-1}$, which was an increase of 44.33% compared with the control. The soil sucrase activity of *T. cuspidata* irrigated with a high concentration (40.0 g·L$^{-1}$) of *F. carica* root extract had no significant difference compared with the control

($p > 0.05$), and was only increased by 3.95%, 2.81%, and 4.76% at 30, 105, and 180 days, respectively, compared with the control ($p < 0.05$) (Figure 6d).

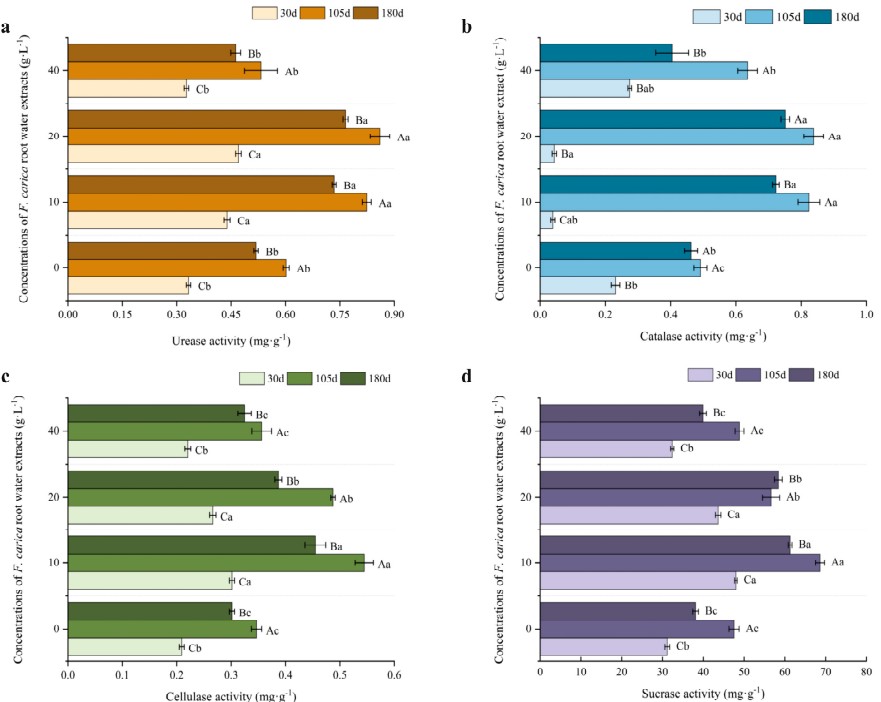

**Figure 6.** The activity of urease (**a**), catalase (**b**), cellulase (**c**), and sucrase (**d**) in the soil of *T. cuspidata* irrigated with water extract of *F. carica* root. Values are reported as mean ± SD, n = 3. Different uppercase letters in the same row indicated significant differences at $p < 0.05$ among different treatments time; Different lowercase letters in the same column indicated significant differences at $p < 0.05$ among different concentration treatments.

### 3.3.3. Effect of Water Extract on Soil Microbial Community

Based on changes in soil microbial diversity, high-throughput sequencing technologies can further reflect changes in bacterial and fungal species diversity and community structure. In the following test, soil samples A1, A2, and A3 were the control (not irrigated with *F. carica* root extract for 30 days, 105 days, and 180 days, respectively). B1, B2, and B3 were irrigated with 10.0 g·L$^{-1}$ *F. carica* root extract for 30 days, 105 days, and 180 days, respectively. C1, C2, and C3 were irrigated with 20.0 g·L$^{-1}$ *F. carica* root extract for 30 days, 105 days, and 180 days, respectively. D1, D2, and D3 were irrigated with 40.0 g·L$^{-1}$ *F. carica* root extract for 30 days, 105 days, and 180 days, respectively.

#### Effects of Different Treatments on α Diversity of Soil Microorganisms

Through changes in the index, Shannon and Simpson's indices represent community variation. The larger Shannon index reflects the higher community species diversity, while the smaller Simpson index reflects the higher species diversity. As can be seen from Table 1A,B, the richness and diversity of soil bacterial and fungal communities of *T. cuspidata* in different irrigation stages showed certain differences. The Shannon index of soil bacteria A2 samples was the highest, followed by C2, indicating that the diversity of the soil bacteria community of *T. cuspidata* was higher after 105 days of irrigation (1 July). The Simpson index of soil bacteria reached its maximum value 30 days after the set concentration of irrigation, indicating that the soil bacterial community diversity of *T. cuspidata* was low 30 days after irrigation (15 April). The Shannon index of soil fungi was the highest after irrigation for 180 days at the set concentration except for 10.0 g·L$^{-1}$. The Simpson index of soil fungi was the lowest after 180 days of irrigating, indicating that the soil fungal community diversity of *T. cuspidata* was higher after 180 days of irrigating (15 September).

**Table 1.** α diversity index of soil bacteria (A) and fungi (B) in *T. cuspidata* (OTU level).

| Sample | Shannon | Simpson | Ace | Chao1 | Coverage |
|--------|---------|---------|-----|-------|----------|
| | | | A: *bacteria* | | |
| A1 | 6.34 ± 0.28 Aa | 0.0057 ± 0.0002 Ab | 3251.70 ± 137.24 Aa | 3237.81 ± 146.74 Aa | 0.9666 ± 0.0013 Aa |
| A2 | 6.54 ± 0.29 Aa | 0.0037 ± 0.0001 Cb | 3293.86 ± 148.76 Aa | 3230.11 ± 151.46 Aa | 0.9672 ± 0.008 Aa |
| A3 | 6.5 ± 0.32 Aa | 0.0041 ± 0.0001 Bc | 3185.11 ± 139.84 Aa | 3164.75 ± 144.11 Aa | 0.9681 ± 0.0012 Aa |
| B1 | 6.25 ± 0.40 Aa | 0.0055 ± 0.0002 Ab | 2953.54 ± 136.49 ABb | 3026.41 ± 153.06 Ab | 0.9696 ± 0.0009 Aa |
| B2 | 6.37 ± 0.39 Ab | 0.0046 ± 0.0001 Ba | 2851.46 ± 142.81 Bb | 2802.97 ± 142.12 Ab | 0.9723 ± 0.0016 Aa |
| B3 | 6.43 ± 0.45 Aa | 0.0043 ± 0.0001 Cbc | 3076.59 ± 140.93 Ab | 3027.90 ± 154.12 Aab | 0.9692 ± 0.0014 Aa |
| C1 | 6.03 ± 0.29 Bb | 0.0091 ± 0.0003 Aa | 2610.63 ± 118.32 Bc | 2526.66 ± 105.80 Bd | 0.9750 ± 0.0005 Aa |
| C2 | 6.47 ± 0.37 Aa | 0.0039 ± 0.0001 Cb | 3060.00 ± 147.33 Ab | 2933.94 ± 128.02 Ab | 0.9700 ± 0.0021 Aa |
| C3 | 6.27 ± 0.32 Ab | 0.0055 ± 0.0002 Ba | 2864.25 ± 133.27 Abc | 2880.64 ± 126.42 Ab | 0.9711 ± 0.0017 Aa |
| D1 | 6.21 ± 0.28 Aa | 0.0051 ± 0.0002 Ac | 2779.35 ± 130.21 Bbc | 2822.53 ± 114.68 Ac | 0.9717 ± 0.0010 Aa |
| D2 | 6.35 ± 0.26Ab | 0.0048 ± 0.0001ABa | 2997.85 ± 142.09Ab | 2949.22 ± 125.48Ab | 0.9701 ± 0.0005Aa |
| D3 | 6.34 ± 0.35 Aa | 0.0045 ± 0.0001 Bb | 2660.82 ± 135.94 Bc | 2642.08 ± 137.26 Bc | 0.9749 ± 0.0014 Aa |
| | | | B: *fungi* | | |
| A1 | 4.84 ± 0.19 Aa | 0.0264 ± 0.0011 Aa | 951.53 ± 38.06 Bb | 953.92 ± 48.16 Bb | 0.9976 ± 0.0009 Aa |
| A2 | 4.67 ± 0.19 Ab | 0.0221 ± 0.0013 ABa | 1253.98 ± 52.16 Aa | 1240.38 ± 59.62 Aa | 0.9946 ± 0.0008 Aa |
| A3 | 4.94 ± 0.20 Aa | 0.0174 ± 0.0007 Ba | 1205.03 ± 48.20 Aa | 1207.17 ± 58.29 Aab | 0.9957 ± 0.0005 Aa |
| B1 | 4.79 ± 0.17 Bab | 0.0242 ± 0.0006 Aa | 1028.49 ± 45.14 Ba | 1026.60 ± 51.06 Ba | 0.9970 ± 0.008 Aa |
| B2 | 5.02 ± 0.18 Aa | 0.0169 ± 0.0012 Ab | 1015.63 ± 43.63 Bb | 1034.56 ± 41.04 Bb | 0.9975 ± 0.0002 Aa |
| B3 | 4.99 ± 0.21 Aa | 0.0174 ± 0.0015 Aa | 1133.12 ± 39.99 Ab | 1123.13 ± 43.69 Ab | 0.9963 ± 0.0007 Aa |
| C1 | 4.89 ± 0.19 Aa | 0.0191 ± 0.0018 Ab | 987.60 ± 37.63 Bab | 996.17 ± 39.89 Cb | 0.9968 ± 0.0009 Aa |
| C2 | 4.94 ± 0.18 Aa | 0.0195 ± 0.0013 Aab | 1050.64 ± 48.52 Bb | 1078.28 ± 42.35 Bb | 0.9970 ± 0.0006 Aa |
| C3 | 5.06 ± 0.20 Aa | 0.0172 ± 0.0005 Ba | 1285.57 ± 58.57 Aa | 1311.17 ± 57.27 Aa | 0.9954 ± 0.0009 Aa |
| D1 | 4.70 ± 0.19 ABb | 0.0233 ± 0.0012Aab | 663.10 ± 40.89 Cc | 668.57 ± 36.74 Cc | 0.9990 ± 0.0001 Aa |
| D2 | 4.60 ± 0.17 Bb | 0.0254 ± 0.0007 Aa | 996.87 ± 39.91 Bb | 987.83 ± 49.51 Bc | 0.9962 ± 0.0003 Aa |
| D3 | 4.98 ± 0.18 Aa | 0.0163 ± 0.0008 Ba | 1154.25 ± 54.63 Ab | 1167.26 ± 56.69 Ab | 0.9961 ± 0.0007 Aa |

Note: Different uppercase letters in the same row indicated significant differences at $p < 0.05$ among different treatments time; Different lowercase letters in the same column indicated significant differences at $p < 0.05$ among different concentration treatments.

Ace and Chao1 indices reflect the richness of the community through the change in the index, and the higher the Ace and Chao1 index, the higher the species richness of the community. The Ace index of soil bacteria was highest at 105 days after irrigation except for 10.0 g·L$^{-1}$. This indicated that the richness of the soil bacterial community of *T. cuspidata* was higher after 105 days of irrigation (1 July). The Chao1 index of soil bacteria peaked at 105 days under 20.0 g·L$^{-1}$ and 40.0 g·L$^{-1}$ irrigation. At 0.0 g·L$^{-1}$ and 10.0 g·L$^{-1}$ of irrigation, the maximum appeared at 30 days and 180 days, respectively. Except for the control, the Ace index and Chao1 index of soil fungi of *T. cuspidata* were the highest at 180 days, indicating that the richness of the soil fungi community of *T. cuspidata* was high at 180 days (15 September). The coverage index reflects the reliability of the sequencing results. The coverage index of soil bacteria is greater than 96%, and the coverage index of soil fungi is greater than 99%, with high coverage, indicating that most bacteria and fungi populations were detected, which can represent the real status of microorganisms in the samples. In summary, *F. carica* root extract had different effects on the diversity and richness of soil bacteria and fungi with different irrigation times. The diversity and richness of the soil bacteria community in *T. cuspidata* soil were higher at 105 days of irrigation (1 July). The effect on the soil fungal community mainly occurred at 180 days of irrigation (15 September).

Effects of Different Treatments on β Diversity of Soil Microorganisms

To investigate the differences in the soil bacterial community structure of *T. cuspidata* between different treatments, nonmetric multidimensional scaling analysis (NMDS) was conducted based on the Bray–Curtis distance. The degree of difference between different regions can be indicated by the distance between points by controlling other variables to examine the response of various treatments to microbial communities. The soil bacterial

communities of *T. cuspidata* irrigated with different concentrations were divided into four different levels by nonmetric multidimensional scaling analysis at the OTU level.

The stress function value = 0.063 < 0.1, which means that the graph has certain explanatory significance and the sample ranking is good. Among them, soil bacterial communities in four strata overlapped partially, and all sample sites were close to relatively clustered, indicating that the species composition of all samples was similar (Figure 7a). In contrast, different sites of soil fungi were more dispersed, and there was a certain distance between samples irrigated at a different time within the same horizon, indicating that there were differences in the composition of the fungi community in the soil samples irrigated at different times. At the same time, ANOSIM was used to analyze the similarity of the soil fungal communities of *T. cuspidata* under different timings of irrigation (Figure 7b). The results showed that there were significant differences in the soil fungal community composition of *T. cuspidata* under different OTU levels considering the phylogenetic evolutionary distance and fungal species abundance (R = 0.6327, *p* = 0.001). These results indicate that *F. carica* root extract could change the soil bacterial community structure of *T. cuspidata*.

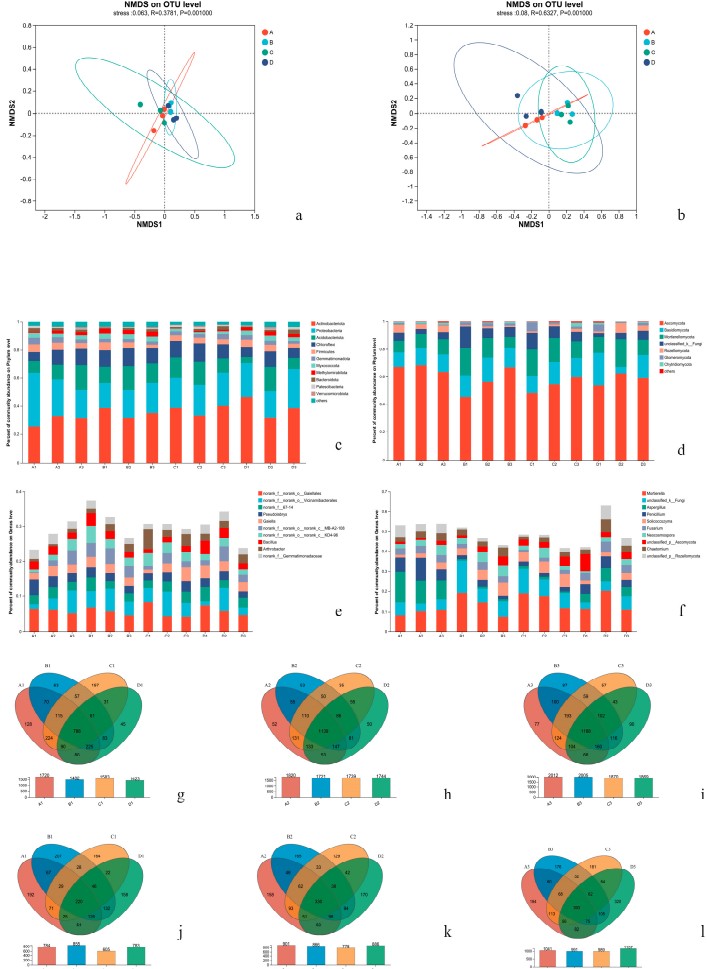

**Figure 7.** β diversity NMDS analysis of bacteria (**a**) and fungi (**b**), community composition analysis of bacteria (**c**) and fungi (**d**) at phylum level and bacteria (**e**) and fungi (**f**) at genus level, species Venn diagram analysis of bacteria (**g**) and fungi (**h**) at 30 days, bacteria (**i**) and fungi (**j**) at 105 days, and bacteria (**k**) and fungi (**l**) at 180 days of *T. cuspidata* from *F. carica* water extract. A1, A2, and A3 were the control (irrigated distilled water for 30, 105 and 180 days, respectively). B1, B2, and B3 were irrigated with 10.0 g·L$^{-1}$ F. carica root extract for 30, 105 and 180 days, respectively. C1, C2, and C3 were irrigated with 20.0 g·L$^{-1}$ F. carica root extract for 30 days, 105 days, and 180 days, respectively. D1, D2, and D3 were irrigated with 40.0 g·L$^{-1}$ F. carica root extract for 30, 105 and 180 days, respectively.

Effects of Different Treatments on Soil Microbial Community Composition and Abundance

As shown in Figure 7c, based on phylum classification level analysis, a total of 33 phyla of bacteria were detected in *T. cuspidata* soil irrigated with different concentrations of *F. carica* root extract after 30 days. Compared with A1, both B1 and D1 contained the deficient GAL15 bacteria, while C1 had the deficient Fibrobacterota and Actinobacteria, and there was an increase in the abundance ratio of Actinobacteria A total of 36 bacterial phyla were detected in *T. cuspidata* soil irrigated with *F. carica* root extract at different concentrations after 105 days. Compared with A2, both B2 and D2 contained the deficient WPS-2 bacteria, while C2 had the deficient Fibrobacterota and the abundance of Acidobacteria increased. A total of 36 bacterial phyla were detected in *T. cuspidata* soil irrigated with *F. carica* root extract at different concentrations during 180 days of irrigation. Compared with A3, B3, C3, and D3 all contained missing Sumerlaeota, Dadabacteria, and Fibrobacterota, while C3 contained missing TX1A-33. The abundance of Actinobacteria and Proteobacteria increased.

Based on phylum classification level analysis, a total of 16 phyla of fungi were detected in *T. cuspidata* soil irrigated with different concentrations of *F. carica* root extract after 30 days. It can be seen from Figure 7d that, compared with A1, both B1 and C1 contained missing Monoblepharomycota, and B1 contained missing Calcarisporiellomycota and Aphelidiomycota. The proportion of Basidiomycota fungi abundance increased. A total of 15 fungal phyla were detected in *T. cuspidata* soil irrigated with *F. carica* root extract at different concentrations after 105 days. Compared with A2, both B2 and D2 contained missing Monoblepharomycota and Calcarisporiellomycota, and the abundance of Mortierellomycota fungi increased. A total of 14 fungal phyla were detected in *T. cuspidata* soil irrigated with *F. carica* root extract at different concentrations during 180 days of irrigation. Compared with A3, the relative abundance of Basidiomycota after irrigation with *F. carica* root extract increased by 1.08%, 0.78%, and 3.96% in B3, C3, and D3, respectively.

Based on the taxonomic analysis, a total of 576 bacteria genera were detected in *T. cuspidata* soil irrigated with *F. carica* root extract at different concentrations after 30 days. As shown in Figure 7e, Arthrobacter was increased by 1.10%, 2.32%, and 1.34% in B1, C1, and D1, respectively, compared with A1. A total of 557 bacteria genera were detected in *T. cuspidata* soil irrigated with *F. carica* root extract at different concentrations for 105 days. Vicinamibacterales had the highest proportion in *T. cuspidata* soil irrigated with different concentrations of *F. carica* root extract. In A2, B2, C2, and D2, these increased by 3.10%, 6.20%, 6.94%, and 6.31%, respectively. A total of 608 bacteria genera were detected in *T. cuspidata* soil irrigated with *F. carica* root extract at different concentrations for 180 days. Compared with A3, the relative abundance of Arthrobacter after irrigation with *F. carica* root extract increased by 0.49%, 4.97%, and 1.12% in B3, C3, and D3, respectively.

A total of 447 fungi genera were detected in *T. cuspidata* soil irrigated with *F. carica* root extract at different concentrations after 30 days. As shown in Figure 7f, Mortierella accounted for the highest proportion in *T. cuspidata* soil irrigated with different concentrations of *F. carica* root extracts. Compared with A1, the abundance proportion of Mortierella fungi in B1, C1, and D1 increased. A total of 430 fungi genera were detected in *T. cuspidata* soil irrigated with *F. carica* root extract at different concentrations after 105 days. Compared with A2, the relative abundance of Mortierella increased in B2, C2, and D2, while the relative abundance of Penicillium decreased. A total of 480 genera of fungi were detected in *T. cuspidata* soil irrigated with *F. carica* root extract at different concentrations for 180 days. Compared with A2, the relative abundance of Penicillium in B2, C2, and D2 decreased by 4.23%, 3.61%, and 2.17%, respectively.

Our results show that the soil bacterial diversity and richness of *T. cuspidata* were the highest at 105 days of *F. carica* root extract irrigation, and the soil fungal diversity and richness of *T. cuspidata* were highest at 180 days of *F. carica* root extract irrigation. After irrigating with *F. carica* root extract, the bacterial community structure of *T. cuspidata* was changed, and the abundance of beneficial microorganisms such as Actinobacteria, Proteobacteria, and Acidobacteria were enriched at the phylum level. The abundance of Arthrobacter beneficial microorganisms was enhanced at the genus level. The fungal

community structure of *T. cuspidata* irrigated with *F. carica* root extract was also different, and the abundance of beneficial microorganisms of Basidiomycota and Mortierellomycota were enriched at the phylum level. At the genus level, the relative abundance of Mortierella was increased, while that of Penicillium was decreased. These dominant microbial groups can produce various extracellular hydrolase, degrade various insoluble organic substances in soil to obtain various nutrients required for cell metabolism, and play an important role in the mineralization of organic matter, to participate in the natural cycles, purify the environment, and enhance soil fertility [38]. At the same time, they resist rhizosphere harmful microorganisms in various ways, inhibit the growth of pathogenic bacteria, to reduce the impact of diseases on plant growth and development, and indirectly promote plant colonization, growth, and development [39].

The species Venn chart is mainly used to calculate the number of operational taxonomic units (OTU) unique to and shared by each sample. On the other hand, it can also intuitively reflect the similarity and overlapping between different samples. Based on OTU levels, it was found that species numbers in soil bacteria of *T. cuspidata* were different after 30 days, 105 days, and 180 days after *F. carica* root extract was irrigated with different concentrations. A1 contained 1720 bacterial OTUs, B1 contained 1482 bacterial OTUs, C1 contained 1583 bacterial OTUs, and D1 contained 1423 bacterial OTUs (Figure 7g). There are 788 OTUs for A1, B1, C1, and D1. A2 contained 1820 bacterial OTUs, B2 contained 1721 bacterial OTUs, C2 contained 1739 bacterial OTUs, and D2 contained 1744 bacterial OTUs. There are 1139 OTUs for A2, B2, C2, and D2 (Figure 7h). A total of 2012 bacterial OTUs were found in A3, 2005 bacterial OTUs were found in B3, 1870 bacterial OTUs were found in C3, and 1869 bacterial OTUs were found in D3 (Figure 7i). In addition, there are 1188 OTUs for A3, B3, C3, and D3. The overall performance was a trend of continuous increase from March to September under the same concentration of irrigation, and the number of OTUs was in the order of 180 days > 105 days > 30 days.

There were differences in the number of species of soil fungi in *T. cuspidata* after 30 days, 105 days, and 180 days of irrigation of different concentrations of *F. carica* root extract. Among them, A1 contained 784 fungal OTUs, B1 contained 855 bacterial OTUs, C1 contained 605 bacterial OTUs, and D1 contained 783 bacterial OTUs. In addition, there are 220 OTUs for A1, B1, C1, and D1 (Figure 7j). A2 had 901 fungal OTUs, B2 had 866 fungal OTUs, C2 had 778 fungal OTUs, and D2 had 886 fungal OTUs (Figure 7k). In addition, there are 330 OTUs for A2, B2, C2, and D2. A total of 1041 fungal OTUs were found in A3, 991 in B3, 989 in C3, and 1157 in D3 (Figure 7l). In addition, there are 393 OTUs for A3, B3, C3, and D3.

By comparing the Venn diagram, it was found that there are differences in the number of bacterial and fungal communities in *T. cuspidata* soil watered by different treatments of *F. carica* root extracts, and they all show a trend of continuous increase from March to September under the same concentration of irrigation. At the same time, the number of endemic communities under $0.0 \ \mathrm{g \cdot L^{-1}}$, $10.0 \ \mathrm{g \cdot L^{-1}}$, and $20.0 \ \mathrm{g \cdot L^{-1}}$ irrigation decreased first and then increased from March to September. The number of endemic communities increased continuously when $40.0 \ \mathrm{g \cdot L^{-1}}$ was irrigated. In terms of the number of unique bacterial communities, $10.0 \ \mathrm{g \cdot L^{-1}}$ was higher than that of the control at 105 days and 180 days. A concentration of $20.0 \ \mathrm{g \cdot L^{-1}}$ was higher than the control at 30 days; $40.0 \ \mathrm{g \cdot L^{-1}}$ was higher than the control at 180 days. In terms of the number of endemic fungal communities, $10.0 \ \mathrm{g \cdot L^{-1}}$ was higher than that of the control at 30 days and 105 days. A concentration of $40.0 \ \mathrm{g \cdot L^{-1}}$ was higher than the control at 105 days and 180 days.

### 3.4. Determination of Potential Growth-Promoting Chemicals in Extracts

UPLC-QTOF-MS/MS was used for the analysis of *F. carica* root water extract and *F. carica* root exudates under Section 2.5. The results showed that the chemicals in the water extract of *F. carica* root had good signals under both positive and negative modes, and were separated well. Combined with the screening results of positive and negative ions, the presence of at least two excimer ions was used as the criterion to confirm the existence of

the screening compounds and lock the target compounds. Based on the literature reports, the secondary mass spectrometry fragment ions of the target compounds were analyzed and compared with the search results of ChemSpider, mzVault, Mass List Search, and other databases (database matching score > 95), to reasonably predict the specific chemical structure of the target compounds. The comparison of chemicals between *F. carica* root water extract and *F. carica* root exudates is detailed in Table 2. The compound type, retention time, main ion peak, target compound, and related information are listed in Table S1 (*F. carica* root water extract) and Table S2 (*F. carica* root exudates). A total of 103 chemicals were detected in the *F. carica* root water extract and *F. carica* root exudates with relative accuracy < 10 ppm, including 12 coumarins, 18 terpenoids, 28 alkaloids, 16 flavonoids, 20 organic acids, 5 steroids, and 5 quinones. There were 62 chemicals shared by *F. carica* root water extract and *F. carica* root exudates, 29 chemicals unique to *F. carica* root water extract, and 13 chemicals unique to *F. carica* root exudates.

**Table 2.** Comparison of chemicals between *F. carica* root water extract and *F. carica* root exudates.

| Type of Compound | RT (min) | Formula | Proposed Compound | *F. carica* Root Water Extract | *F. carica* Root Exudates |
|---|---|---|---|---|---|
| Coumarins | 3.94 | $C_9H_6O_4$ | Esculetin | + | + |
| | 4.88 | $C_{11}H_6O_4$ | Xanthotoxol | + | + |
| | 4.89 | $C_{11}H_6O_4$ | Bergaptol | + | + |
| | 5.57 | $C_9H_6O_3$ | Umbelliferone | + | + |
| | 6.77 | $C_{14}H_{12}O_3$ | Angenomalin | + | + |
| | 6.83 | $C_{14}H_{12}O_3$ | Trioxsalen | + | − |
| | 8.17 | $C_{14}H_{14}O_4$ | Nodakenetin | + | + |
| | 8.26 | $C_{14}H_{14}O_4$ | Columbianetin | + | + |
| | 9.63 | $C_{11}H_6O_3$ | Psoralen | + | + |
| | 11.21 | $C_{12}H_8O_4$ | Bergapten | + | + |
| | 25.70 | $C_{21}H_{24}O_5$ | Rutamarin | + | + |
| | 35.59 | $C_{11}H_{10}O_3$ | 7- Ethoxycoumarin | + | + |
| Terpenoids | 0.73 | $C_{13}H_{16}N_4O_6$ | Furaltadone | + | + |
| | 0.75 | $C_3H_6N_6O_5$ | 1,3-Dinitro-5-nitroso-1,3,5-triazinane | + | - |
| | 3.66 | $C_{20}H_{24}O_{10}$ | Ginkgolide B | − | + |
| | 4.22 | $C_{11}H_{15}N_3O_8$ | Polyoxin C | − | + |
| | 5.61 | $C_{31}H_{32}N_4O_9$ | Cotarnine phthalate | + | − |
| | 7.23 | $C_{25}H_{22}N_2O_5$ | Zanamivir Amine Triacetate Methyl Ester | + | + |
| | 10.57 | $C_{28}H_{22}$ | 9,10-Bis(4-methylphenyl)anthracene | + | + |
| | 11.20 | $C_9H_3N_3O_3$ | 1,3,5-Triisocyanatobenzene | + | − |
| | 15.30 | $C_{26}H_{26}O_9$ | Gilvocarcin M | + | + |
| | 19.36 | $C_{20}H_{22}O_2$ | Longistylin | − | + |
| | 24.10 | $C_{14}H_{26}O_4$ | Tetradecaneioic acid | − | + |

**Table 2.** *Cont.*

| Type of Compound | RT (min) | Formula | Proposed Compound | *F. carica* Root Water Extract | *F. carica* Root Exudates |
|---|---|---|---|---|---|
| | 26.91 | $C_{38}H_{49}N_3O_5$ | Bemotrizinol | + | + |
| | 27.27 | $C_{24}H_{38}O_4$ | Dioctyl phthalate | + | + |
| | 27.76 | $C_{25}H_{35}NO_4$ | Norbuprenorphine | + | + |
| | 29.56 | $C_{30}H_{46}O_3$ | Wilforlide A | + | − |
| | 31.46 | $C_{15}H_{22}O_4$ | Zinniol | + | + |
| | 31.94 | $C_{12}H_{16}O$ | 4-Cyclohexylphenol | − | + |
| | 34.61 | $C_{17}H_{26}O_5$ | Decyl Gallate | + | + |
| Alkaloids | 0.69 | $C_{11}H_6O_6$ | 4-Azido-3,5-dinitrobenzonitrile | + | − |
| | 0.82 | $C_{14}H_{21}NO_3$ | 2,6-ditert-butyl-4-nitrophenol | − | + |
| | 0.90 | $C_{21}H_{16}O_6$ | Justicidin B | + | + |
| | 0.93 | $C_4H_2N_6O_5$ | Azoxyfuroxan | + | + |
| | 1.22 | $C_{13}H_{20}N_2O_6$ | 5-Butyluridine | − | + |
| | 4.28 | $C_{22}H_{26}O_{12}$ | Catalposide | + | − |
| | 4.98 | $C_{32}H_{18}N_8$ | Phthalocyanine | + | − |
| | 5.55 | $C_{20}H_{24}O_{10}$ | 4-[4-(3-Phenyl-2-quinoxalinyl)phthalonitrile | + | + |
| | 5.93 | $C_{30}H_{22}N_4O_4$ | Adozelesin | + | − |
| | 6.07 | $C_{15}H_{18}O_4$ | Helenalin | + | + |
| | 7.15 | $C_{19}H_{20}N_6O_6$ | Azido-Thalidomide | + | − |
| | 7.30 | $C_{18}H_{26}N_2O_9$ | Nicametate citrate | + | − |
| | 7.38 | $C_{37}H_{28}O_{10}$ | Pyranoamentoflavone | + | + |
| | 8.11 | $C_{18}H_{22}N_2O_{10}$ | Nitrophenyl 2-acetamido-3,6-di-O-acety-2-deoxyl-beta-D-glucopyranoside | + | + |
| | 13.89 | $C_{19}H_{38}N_2O_3$ | N-Pentadecyl-L-asparagine | + | − |
| | 13.94 | $C_{16}H_{35}NO_2$ | Lauryldimethylammonium Acetate | + | + |
| | 14.03 | $C_{14}H_{31}NO$ | Lauryldimethylamine oxide | + | + |
| | 16.29 | $C_{18}H_{39}NO_2$ | Safingol | + | + |
| | 17.32 | $C_{15}H_{18}N_6O_2$ | Pimefylline | − | + |
| | 17.51 | $C_{21}H_{37}N$ | N,N-Dimethylandrostan-1-amine | + | + |
| | 19.60 | $C_7H_{17}N$ | N,N-dimethylpentan-3-amine | + | + |
| | 20.39 | $C_{22}H_{47}NO_2$ | Ammonium Behenate | + | + |
| | 20.65 | $C_{22}H_{47}N$ | Docosylamine | + | − |
| | 23.36 | $C_{18}H_{35}NO$ | 9-Octadecenamide | + | − |
| | 25.33 | $C_{18}H_{37}NO$ | Amide C18 | + | + |
| | 25.58 | $C_{20}H_{41}NO$ | N,N-Dimethylstearamide | + | + |
| | 35.14 | $C_{19}H_{30}O_5$ | Dodecyl gallate | + | + |

**Table 2.** *Cont.*

| Type of Compound | RT (min) | Formula | Proposed Compound | *F. carica* Root Water Extract | *F. carica* Root Exudates |
|---|---|---|---|---|---|
| Flavones | 0.70 | $C_{30}H_{22}O_{10}$ | Isochamaejasmin | + | − |
| | 0.85 | $C_{30}H_{26}O_{13}$ | Proanthocyanidins | + | + |
| | 0.92 | $C_{15}H_{12}O_7$ | Taxifolin | + | + |
| | 2.49 | $C_{15}H_{14}O_6$ | (-)-Epicatechin | + | + |
| | 3.16 | $C_{30}H_{26}O_{12}$ | Procyanidin B2 | + | + |
| | 3.31 | $C_{21}H_{23}N_3O_7$ | Amicycline | + | + |
| | 3.91 | $C_{30}H_{26}O_{11}$ | Gambiriin C | + | + |
| | 4.45 | $C_{30}H_{26}O_{12}$ | Procyanidin B3 | + | + |
| | 4.47 | $C_{26}H_{28}O_{15}$ | Neocarlinoside | + | − |
| | 4.92 | $C_{24}H_{22}O_{13}$ | Malonylgenistin | + | + |
| | 7.25 | $C_{28}H_{34}O_{15}$ | Hesperidin | + | + |
| | 8.86 | $C_{15}H_{11}O_6$ | Cyanidin cation | + | + |
| | 23.99 | $C_{27}H_{30}O_{16}$ | Rutin | − | + |
| | 35.14 | $C_{19}H_{30}O_5$ | Dodecyl gallate | + | − |
| | 35.56 | $C_{27}H_{32}O_{14}$ | Naringin | + | + |
| | 35.60 | $C_{28}H_{34}O_{15}$ | Neohesperidin | + | + |
| Organic acids | 0.71 | $C_{12}H_{21}N_3O_6$ | Boc-Ala-Gly-Gly | + | − |
| | 0.94 | $C_{11}H_{20}N_2O_3$ | Pro-leu | + | − |
| | 0.95 | $C_{10}H_{13}NO_2$ | Phenprobamate | + | + |
| | 1.22 | $C_5H_{11}NO_2$ | Valine | + | − |
| | 4.51 | $C_{11}H_{18}N_2O_2$ | Cyclo(leucylprolyl) | + | + |
| | 5.21 | $C_{10}H_{19}N_7O_5$ | neosaxitoxin conjugate acid | − | + |
| | 5.34 | $C_{24}H_{39}N_7O_8$ | L-Threonyl-L-histidyl-L-threonyl-L-valyl-L-proline | + | + |
| | 6.43 | $C_9H_{14}N_2O_7$ | Asp-Glu | + | − |
| | 9.36 | $C_6H_{12}N_2O_4$ | Glycyl-D-threonine | + | + |
| | 11.10 | $C_{10}H_5O_3$ | Oxo-4-phenyl-3-butynoate | + | + |
| | 11.32 | $C_{24}H_{20}O_{10}$ | Gyrophoric acid | + | − |
| | 13.35 | $C_{10}H_{16}N_2O_8$ | Ethylenediaminetetraacetic acid | + | + |
| | 19.67 | $C_{24}H_{40}O_2$ | 5β-cholanoic acid | + | + |
| | 21.11 | $C_{14}H_{26}N_2O_6$ | BOC-L-DAB(BOC) | − | + |
| | 27.78 | $C_{19}H_{38}N_2O_6$ | 2,3-dihydroxybutanedioic acid | + | + |
| | 30.62 | $C_{15}H_{22}O_3$ | gemfibrozil | + | + |
| | 32.15 | $C_{44}H_{58}N_2O_3$ | 4-[28-Oxo-28-{[2-(2-pyridinyl)ethyl]amino}lupa-2,20(29)-dien-3-yl]benzoic acid | + | + |

**Table 2.** *Cont.*

| Type of Compound | RT (min) | Formula | Proposed Compound | *F. carica* Root Water Extract | *F. carica* Root Exudates |
|---|---|---|---|---|---|
| | 32.22 | $C_{39}H_{58}N_4O_5$ | 3-hexanoyl-NBD Cholesterol | + | + |
| | 32.34 | $C_{26}H_{52}O_3$ | 26-Hydroxyhexacosanoic acid | + | − |
| | 33.68 | $C_6H_8O_7$ | Citric Acid | + | − |
| | 0.74 | $C_{11}H_{15}N_5O_3$ | Lobucavir | + | − |
| | 4.63 | $C_{22}H_{14}$ | Pentaphene | + | − |
| Steroids | 5.77 | $C_{14}H_7NO_5$ | 1-Hydroxy-3-nitro-9,10-anthraquinone | − | + |
| | 15.37 | $C_{24}H_{30}O_6$ | Meprednisone Acetate | + | + |
| | 27.63 | $C_{19}H_{26}O_6$ | Arctiopicri | − | + |
| | 5.20 | $C_{32}H_{28}O_9$ | Shikometabolin B | + | − |
| | 6.66 | $C_{27}H_{30}O_{13}$ | Lignan P | + | − |
| Quinones | 32.18 | $C_{17}H_{26}O_4$ | Embelin | + | + |
| | 35.53 | $C_{19}H_{30}O_4$ | Decylubiquinone | + | + |
| | 35.63 | $C_{11}H_8O_3$ | Plumbagin | + | + |

Note: "+" means the chemical is present and "−" means the chemical is not present.

## 4. Discussion

At the same irrigation time, the plant height and base diameter of *T. cuspidata* irrigated with different concentrations of *F. carica* root extract increased first and then decreased with the increase in *F. carica* root extract concentration (Figure 1). All of them showed the trend of "promoting low and suppressing high". Only from the view of plant height and base diameter of *T. cuspidata*, was *F. carica* root extract with 10.0 g·L$^{-1}$ concentration superior to other concentrations. This is consistent with the results of other studies on the concentration effect of "low promotion and high inhibition" in plant interaction [5,40]. The reason for the weakening of the promoting effect of high-concentration irrigation may be that the plant interaction becomes stronger when the concentration of *F. carica* root extract increases, while the physiological morphology of the young roots of *T. cuspidata* has not been fully formed, and the protective enzyme system and defense system in the plant are also poor, so they are more sensitive to plant metabolites and other external stimuli, and are more prone to disease and adversity [27,41,42].

Figure 2 shows the photosynthetic results of *T. cuspidata* irrigated with *F. carica* root extract of different concentrations. Compared with the control, a low concentration (10.0 g·L$^{-1}$) and medium concentration (20.0 g·L$^{-1}$) of *F. carica* root extract significantly promoted the contents of Pn, Tr, Gs, and Ci. These results indicate that a lower concentration of *F. carica* root extract irrigation can promote the stomatal opening of *T. cuspidata* leaves, enhance transpiration activity, provide sufficient photosynthetic raw materials in cells, improve the efficiency of light energy capture, and thus increase dry matter accumulation [43]. The decrease in the contents of Pn, Tr, Gs, and Ci in *T. cuspidata* leaves irrigated with a high concentration of 40.0 g·L$^{-1}$ *F. carica* root extract may be due to the inhibition of the activity of ATPase in plant cell mitochondria, which reduced the respiration rate of plants and reduced the photosynthetic performance [44]. It may also be that the high-concentration irrigation reduced the self-regulation ability of *T. cuspidata* and damaged the $CO_2$ transfer pathway from the stomatal cavity to the chloroplast carboxylation site, resulting in the reduction in the $CO_2$ transport rate, and affected the photosynthetic electron transport chain in *T. cuspidata* leaves [45].

With the increase in *F. carica* root extract concentration at the same irrigation time, the contents of chlorophyll a, chlorophyll b, and total chlorophyll in the leaves of *T. cuspidata* showed a trend of first rising and then falling (Figure 3). The contents of chlorophyll a, chlorophyll b, and total chlorophyll in the leaves of *T. cuspidata* increased under lower concentrations of *F. carica* root extracts (10.0 g·L$^{-1}$ and 20.0 g·L$^{-1}$), which may be due to the strong regulation ability of *T. cuspidata* and its strong chlorophyll biosynthesis function [46]. The suppression of chlorophyll synthetase activity may be the cause of the decrease in chlorophyll a, chlorophyll b, and total chlorophyll levels in the leaves of *T. cuspidata* watered with a high concentration of *F. carica* root extract (40.0 g·L$^{-1}$), which leads to the change in the pigment protein structure on the thylakoid membrane or the destruction of the chloroplast structure, which leads to chlorophyll degradation [47].

As can be seen in Figure 4, after the *F. carica* root extract was irrigated at a lower concentration, the contents of protective enzymes in *T. cuspidata* leaves increased, while the content of MDA significantly decreased. It was consistent with the changing trend of protective enzymes in *T. cuspidata* leaves. It may be that in the early stage of *T. cuspidata* leaves, oxidizing substances accumulated continuously, and protective enzyme activities were activated and reduced MDA contents [48]. However, under a high concentration of *F. carica* root extract, the activity of the protective enzyme in the leaves of *T. cuspidata* was inhibited and the content of MDA was increased, indicating that high-concentration irrigation significantly inhibited *T. cuspidata* growth, causing a free radical reaction and lipid peroxidation, and damaging the physiological integrity of plant cells. The reason may be that with the increase in the concentration of *F. carica* root extract, the secondary metabolites in the extract also increased, and the stress effect on *T. cuspidata* was enhanced. In the adverse environment, the accumulation of reactive oxygen species in *T. cuspidata* caused serious membrane lipid peroxidation in the leaves of *T. cuspidata,* which affected the cell structure, increased the relative permeability of the plasma membrane and the accumulation of oxidizing substances in the plant body, eventually leading to membrane peroxidation [49].

As can be seen from Figure 5, soil pH, SOC, AN, AP, and AK contents were all increased under the irrigation of *F. carica* root extract at a low concentration (10.0 g·L$^{-1}$) and medium concentration (20.0 g·L$^{-1}$), and showed a trend of first increasing and then decreasing with the increase in concentration. These results indicate that the extract of *F. carica* root might contain antibacterial substances, which could improve the microecological environment and physical and chemical properties of *T. cuspidata* soil in Northeast China. The content of root exudates further increased the soil carbon source, stimulated the soil microbial activity of *T. cuspidata*, enhanced the decomposition of soil organic matter, increased the content of soil organic matter, and promoted the fixation, conversion, and availability of soil nitrogen [50–52]. The activity of soil phosphorus was also enhanced, promoting its conversion into AP. At the same time, under the action of microorganisms, the potassium mineralization and decomposition in the soil of *T. cuspidata* were accelerated, and finally, the content of potassium in the soil was increased [53,54]. The soil EC increased first and then decreased with the irrigation time, and reached the peak value in summer (105 days). This may be due to the high temperature in July, which leads to increased evaporation of water, increased salt content, and higher soil EC. In September, when the temperature drops, the water evaporates less, the salt content decreases, and the soil EC decreases [55].

It can be seen in Figure 6 that the low concentration (10.0 g·L$^{-1}$) and medium concentration (20.0 g·L$^{-1}$) *F. carica* root extract had the most significant promoting effect on the soil enzyme activity of *T. cuspidata*. The reason may be that the nitrogen fixation of *F. carica* root extract at a lower concentration increased the contents of soil organic matter, and the organic matter contained in *F. carica* root extract promoted the growth and propagation of soil microorganisms [56]. The soil enzyme activity was also improved. It is also possible that a lower concentration of *F. carica* root extract can accelerate the decomposition of hydrogen peroxide, which is harmful to plants, to alleviate the damage of *T. cuspidata* to some extent [57]. However, the high concentration of 40.0 g·L$^{-1}$ *F. carica* root extract

inhibited the soil enzyme activities of *T. cuspidata* most significantly, which may be due to the excessive chemicals contained in the high concentration of *F. carica* root extract, which formed a certain inhibition effect on plant growth and led to the decreased in soil enzyme activities [58].

In terms of the mechanism of allelopathy, this study started from the interaction between *T. cuspidata,* soil, and microorganisms mediated by secretions and simulated the potential allelopathy effect of *T. cuspidata* root exudates on *T. cuspidata* with *F. carica* root water extract. Different irrigation times had different effects on the diversity and richness of soil bacteria and fungi. The diversity and richness of the soil bacterial community were higher at 105 days of irrigation, while the influence of the soil fungal community mainly occurred at 180 days of irrigation (Table 1). The bacterial and fungal community structure of *T. cuspidata* was changed by irrigation with *F. carica* root water extract (Figure 7). At the phylum and genus levels, the bacterial community was enriched in plant growth, promoting rhizobacteria, including Actinobacteria, Proteobacteria, Acidobacteria, and Arthrobacter. The bacterial colonies have a positive effect on plants, and can inhibit the growth of pathogenic bacteria, participate in nutrient cycling in plants and secrete plant growth regulators, and promote the absorption and utilization of mineral nutrients in plants [59]. Basidiomycota, Mortierellomycota, and Mortierella were dominant in the soil. They have antagonistic effects on a variety of pathogenic fungi and can effectively degrade refractory organic matter such as lignin and keratins in soil. At the same time, the relative abundance of the pathogen Penicillium in soil decreased after irrigation with *F. carica* root water extract. The action mechanism of beneficial rhizosphere microorganisms is relatively complex. On the one hand, it may be that these growth-promoting bacteria change the physical and chemical properties of soil and the form of ineffective mineral elements in soil, making them more easily absorbed and utilized by *T. cuspidata*, thus directly promoting the growth of *T. cuspidata* [60]. On the other hand, plants can communicate with soil through root exudates, which provide energy for soil microorganisms, change the soil microbial community structure, and improve the population of beneficial microorganisms while restricting the growth of pathogenic bacteria. It also indirectly promotes the growth of *T. cuspidata* [61].

By UPLC-QTOF-MS/MS analysis, the compounds identified in *F. carica* root water extract and *F. carica* root exudates were of the same species and a similar composition, and most of these chemicals were secondary metabolites of *F. carica* [62]. This mainly includes coumarins, terpenoids, alkaloids, flavonoids, organic acids steroids, and quinones (Table 2). Therefore, it is speculated that the enrichment of potential chemicals secreted by *F. carica* root interacts with soil microorganisms of *T. cuspidata*, and increases the relative abundance of growth-promoting bacteria such as Actinobacteria, Proteobacteria, Acidobacteria, Arthrobacter, Basidiomycota, Mortierellomycota, and Mortierella. In addition, beneficial microorganisms competed with pathogenic bacteria in nutrition and site competition, which reduced the growth of the pathogenic bacteria Penicillium, and then promoted the growth and development of *T. cuspidata*. The allelopathic-promoting effect of *F. carica* on *T. cuspidata* was mainly caused by the accumulation of potential chemicals in *F. carica* root exudates in the soil through the interaction with soil microorganisms. Therefore, this is the result of the comprehensive action of chemicals and rhizosphere soil microorganisms.

## 5. Conclusions

This study systematically investigated the effects of *F. carica* root water extract on the growth, physiology, and biochemistry, and the rhizosphere microbial community diversity of *T. cuspidata*. It was found that *F. carica* rhizosphere soil water extract and root exudates had to promote the growth of *T. cuspidata,* and had a great impact on the rhizosphere microbial community structure of *T. cuspidata*. The diversity of microbial communities was changed, the number of growth-promoting bacteria increased, and the number of pathogenic bacteria decreased. A total of 103 chemicals were identified from *F. carica* root water extract and *F. carica* root exudates by UPLC-QTOF-MS/MS, among which

62 chemicals were common to the two. Therefore, it can be preliminarily inferred that *F. carica* roots release a large number of chemicals into the soil during the mixed planting process of *F. carica* and *T. cuspidata*, which may directly promote the growth of *T. cuspidata* by affecting the soil microbial community structure and changing the soil environment.

**Supplementary Materials:** The following supporting information can be downloaded at: https://www.mdpi.com/article/10.3390/f14061213/s1, Table S1. Chemicals in *F. carica* root water extract; Table S2. Chemicals in *F. carica* root exudates.

**Author Contributions:** Q.L.: data curation, writing—original draft preparation, and investigation. J.H.: visualization and investigation. X.Y.: resources and investigation. Z.G.: writing—review and editing. W.S.: formal analysis. B.Q.: resources. J.C.: software and validation. C.L.: conceptualization and supervision. C.Z.: supervision and project administration. All authors have read and agreed to the published version of the manuscript.

**Funding:** This work was supported by the Fundamental Research Funds for the Central Universities (No. 2572022DP06), the Natural Science Foundation of Heilongjiang Province (LH2022C004), the Heilongjiang Touyan Innovation Team Program (Tree Genetics and Breeding Innovation Team), and the Higher Education Discipline Innovation Project (the 111 Project), China (B20088).

**Institutional Review Board Statement:** Not applicable.

**Informed Consent Statement:** Not applicable.

**Data Availability Statement:** The data presented in this study are available on request from the corresponding author.

**Conflicts of Interest:** The authors declare no conflict of interest.

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
