# Peer review of "Effects of Ficus carica L. Water Extract on Taxus cuspidata Sieb. et Zucc. Growth"

_forests, doi:10.3390/f14061213_

Round 1
Reviewer 1 Report
Review report: forests-2421472
The findings could be interesting for researchers. However, following comments should be addressed before proceeding this manuscript for further. Authors are highly recommended to correct manuscript as per following suggestions for enhancing readability and reproducibility of results.
1. Keywords should be arranged in alphabetical order.
2. Introduction is too long.
3. The necessity and innovation of the article should be presented in the last paragraph of introduction section.
4. The following references maybe helpful for this paper and recommended to be cited.
https://doi.org/10.3390/molecules27248999; https://doi.org/10.1002/imt2.66; https://doi.org/10.1016/j.ecolind.2021.108031; https://doi.org/10.1016/j.heliyon.2022.e09094
3. In general, legends of figures and Tables are not all self-explainable. I am recommending that figures and Tables must be self-explanatory. That is, all statistics and abbreviations used must be clearly explained. For example, you should define what you are showing in capital and lowercase in the mean comparison.
4. Please describe the greenhouse situation used in this study.
5. How many replicates have been done for each assay. Please indicate it in the related sections.
6. Section 2.3.4, 2.3.5, 2.3.6, 2.4.2 and 2.4.3, I do not think these methods were for the mentioned references. Have those authors modified the main methods? Please use the main references for these sections.
7. Authors are highly recommended to rearrangement of discussion section like results and vice versa. It will be better to the reader to following the contents.
Author Response
Dear reviewer,
We would like to thank the editor for giving us a chance to revise the artical, and also thank the reviewers for giving us constructive suggestions which would help us both in English and in depth to improve the quality of the artical. Here the version of our artical has been modified according to the editor and reviewers’ suggestions. In addition, Associate Professor Chunying Li was finally identified as the corresponding author.
Sincerely yours,
Chunying Li, Associate Professor
***********************************************************
The following is a point-to-point response to the editor and two reviewers’ comments.
------------------------------------------------------------------------------------
Reviewer #1:
The findings could be interesting for researchers. However, following comments should be addressed before proceeding this manuscript for further. Authors are highly recommended to correct manuscript as per following suggestions for enhancing readability and reproducibility of results.
- Keywords should be arranged in alphabetical order.
>> I have arranged the keywords in alphabetical order (see keywords in the revised manuscript).
- Introduction is too long.
>> The introduction has been optimized and improved (see Introduction in the revised manuscript).
- The necessity and innovation of the article should be presented in the last paragraph of introduction section.
>> The necessity and innovation of the article have been presented (see the last paragraph of introduction section in the revised manuscript).
- The following references maybe helpful for this paper and recommended to be cited.
https://doi.org/10.3390/molecules27248999
https://doi.org/10.1016/j.ecolind.2021.108031
>> We think that these two references are related to our research content and have been quoted as [10] and [52] respectively (see line 70-73 and line 800-803 in the revised manuscript).
- In general, legends of figures and Tables are not all self-explainable. I am recommending that figures and Tables must be self-explanatory. That is, all statistics and abbreviations used must be clearly explained. For example, you should define what you are showing in capital and lowercase in the mean comparison.
>> I've defined what the upper and lower case letters say, see line 322-324 in the revised manuscript. "The same below" in line 325 means that the upper and lower case letters in the chart below in our article have the same meaning.
- Please describe the greenhouse situation used in this study.
>> The greenhouse situation used in this study has been supplemented (see line 119-124 in the revised manuscript).
- How many replicates have been done for each assay. Please indicate it in the related sections.
>> As shown in the relevant section, each experiment was repeated 3 times (see line 161 and 166-167 under section 2.2.3 and line 179-181 under section 2.3.1 in the revised manuscript).
- Section 2.3.4, 2.3.5, 2.3.6, 2.4.2 and 2.4.3, I do not think these methods were for the mentioned references. Have those authors modifiedthe main methods? Please use the main references for these sections.
>> Our experimental methods are based on minor modifications of the references, and the main references in this section have been quoted (see section 2.3.4, 2.3.5, 2.3.6, 2.4.2 and 2.4.3 in the revised manuscript).
- Authors are highly recommended to rearrangement of discussion section like results and vice versa. It will be better to the reader to following the contents.
>>We have rearranged the discussion section according to the content and made appropriate corrections (see line 736- 866 in the revised manuscript).
Reviewer 2 Report
''Promoting effects of Ficus carica L. water extract on Taxus cuspidata Sieb. et Zucc. growth'' is a complex and suggestive title. The abstract is done properly, but I suggest a phrase with the purpose/importance of such a study more clearly defined.
The work as a whole is well done, the introduction is complex. Regarding the material and method, they were well described, and the results are well processed statistically.
There are small errors, such F. carica or T. cuspidata must be written uniformly and with italics... and I also recommend a very short linguistic check.
The conclusions are thoroughly supported by the results presented in the paper. Complete and current bibliography.
I recommend a very short linguistic check.
Author Response
Dear reviewer,
We would like to thank the editor for giving us a chance to revise the artical, and also thank the reviewers for giving us constructive suggestions which would help us both in English and in depth to improve the quality of the artical. Here the version of our artical has been modified according to the editor and reviewers’ suggestions. In addition, Associate Professor Chunying Li was finally identified as the corresponding author.
Sincerely yours,
Chunying Li, Associate Professor
***********************************************************
The following is a point-to-point response to the editor and two reviewers’ comments.
------------------------------------------------------------------------------------
Reviewer #2:
- ''Promoting effects of Ficus carica water extract on Taxus cuspidata Sieb. et Zucc. growth'' is a complex and suggestive title.
>> The title has been revised (see title in the revised manuscript).
- The abstract is done properly, but I suggest a phrase with the purpose/importance of such a study more clearly defined.
>> A phrase has been used to define more clearly the purpose/importance of the study (see abstract in the revised manuscript).
- The work as a whole is well done, the introductionis complex. Regarding the material and method, they were well described, and the results are well processed statistically.
>> The introduction has been optimized and improved (see introduction in the revised manuscript).
- There are small errors, such caricaor T. cuspidata must be written uniformly and with italics... and I also recommend a very short linguistic check.
>> We have modified and checked the language of the whole text in the revised manuscript.
- The conclusions are thoroughly supported by the results presented in the paper. Complete and current bibliography.
>> Thank you very much for your review and comments.
Round 2
Reviewer 1 Report
Dear editor,
Thanks for inviting me to re-evaluate this paper. From my point of view it is ready to be accepted.